# What is Your Data Worth to GPT?
# LLM-Scale Data Valuation with Influence Functions

Sang Keun Choe[1][*] Hwijeen Ahn[1][†] Juhan Bae[2][†] Kewen Zhao[1][†]
Minsoo Kang[3] Youngseog Chung[1] Adithya Pratapa[1] Willie Neiswanger[4]
Emma Strubell[1] Teruko Mitamura[1] Jeff Schneider[1] Eduard Hovy[1] Roger Grosse[2] Eric Xing[1,5]
[1] Carnegie Mellon University   [2] University of Toronto   [3] Georgia Tech   [4] USC   [5] MBZUAI

## Abstract

Large language models (LLMs) are trained on a vast amount of human-written data, but data providers often remain uncredited. In response to this issue, data valuation (or data attribution[2]), which quantifies the contribution or value of each data to the model output, has been discussed as a potential solution. Nevertheless, applying existing data valuation methods to recent LLMs and their vast training datasets has been largely limited by prohibitive compute and memory costs. In this work, we focus on influence functions, a popular gradient-based data valuation method, and significantly improve its scalability with an efficient gradient projection strategy called LOGRA that leverages the gradient structure in backpropagation. We then provide a theoretical motivation of gradient projection approaches to influence functions to promote trust in the data valuation process. Lastly, we lower the barrier to implementing data valuation systems by introducing LOGIX, a software package that can transform existing training code into data valuation code with minimal effort. In our data valuation experiments, LOGRA achieves competitive accuracy against more expensive baselines while showing up to $6{,}500\times/5\times$ improvements in compute/memory efficiency in influence computations as well as $2\times$ speed-up in gradient statistics logging when applied to Llama3-8B-Instruct and the 1B-token subset of the OpenWebText dataset.

## 1 Introduction

Despite the well-recognized importance of training data in advancing the capabilities of large language models (LLMs) [7, 32, 48], there is no agreed-upon mechanisms for crediting or compensating data providers. As LLMs are increasingly integrated into our society and economy, the absence of such mechanisms has aggravated a tension between data and model providers, exemplified by recent legal challenges involving major tech companies [30, 40]. In this atmosphere, data valuation, which quantifies the contribution of each training data to the model output, has been discussed as a potential technical solution for tackling these societal issues [13, 15, 26, 29, 57, 63].

At a high level, most data valuation algorithms interpret the model output as a coalition of its training data, and evaluate the contribution of each example based on its influence on the model output when included or excluded from the training dataset [15, 27, 33, 36]. If an inclusion of a specific training example consistently improves model performance, high value can be assigned to this example for its contribution. However, applying existing data valuation methods to recent LLMs and their vast training datasets has faced significant scalability challenges to date. For instance, sampling-based methods, such as the Shapley value [15, 36] or Datamodels [27], require retraining the model multiple

---

[*]Lead author: sangkeun00@gmail.com.   [†]Main contributors.

[2]Noting that the leave-one-out error [33], a basis for most data attribution methods, is a *semivalue* [10, 36, 54], we use "data valuation" as a unified term in this work.

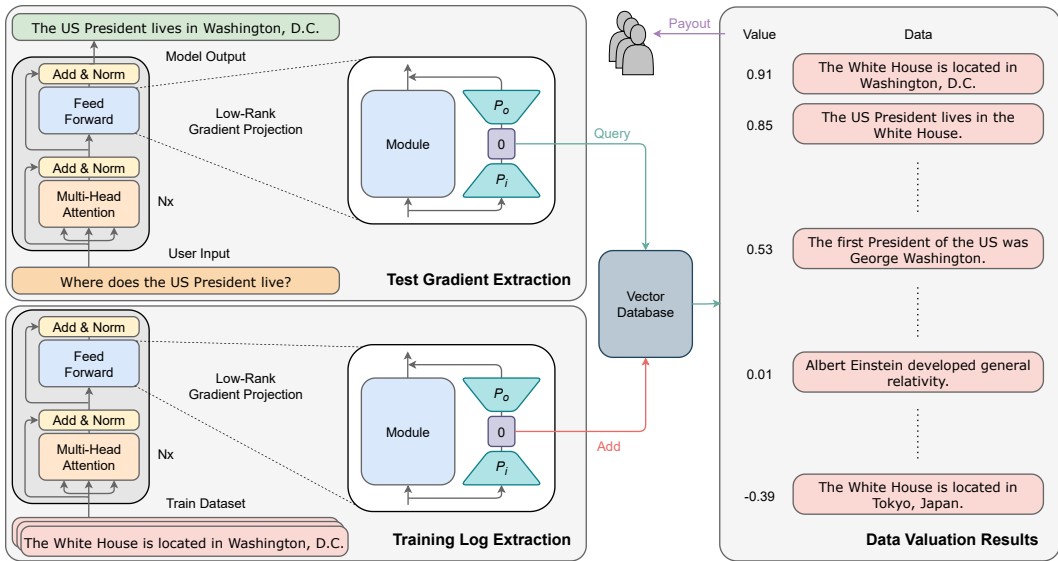

Figure 1: Data valuation system architecture. **(Left Bottom)** We first extract the Hessian and gradients for all training data using efficient gradient projection LoGra and store them in a database. **(Left Top)** At test time, we similarly extract gradients and query the database. **(Right)** The database returns similarity scores with respect to training examples that can be used for data valuation/attribution.

times with varied combinations of data subsets to directly model the effect of in/excluding each data. Unfortunately, such repeated retraining is hardly affordable even for small models, let alone LLMs. To overcome this issue, gradient-based methods, including influence functions [33, 43], approximate the effect of data in/exclusion on the model output using gradient information without costly retraining. Even so, scaling gradient-based methods to LLMs is hindered by prohibitive compute and memory costs originating in the high-dimensional nature of the gradient.

Consequently, the main objective of this work is to bridge the gap in scaling existing data valuation methods to recent LLMs and their vast training datasets. Toward this goal, we focus on influence functions [33, 43], a representative gradient-based data valuation method, and significantly improve its scalability with an efficient gradient projection algorithm. We visualize the proposed data valuation system in Figure 1, and detail our technical contributions below:

- Employing gradient structures in backpropagation, we develop a novel **lo**w-rank **gra**dient projection algorithm LoGra that improves space & time complexity of gradient projection, a major scalability bottleneck in prior work [43, 49], from $O(nk)$ to $O(\sqrt{nk})$ where $n$ and $k$ are model and projection dimensions. Furthermore, LoGra directly computes projected gradients without materializing full gradients, enabling low GPU memory and high GPU utilization for improved efficiency. Lastly, we show that LoGra can be easily implemented with small add-on layers, similarly to LoRA [24].

- By interpreting a damping term in influence functions as a spectral gradient sparsification mechanism, we (1) offer a theoretical motivation of gradient projection approaches to influence functions and (2) derive a specialized PCA initialization scheme for LoGra.

- We introduce software named Logix that (1) makes it *simple* to convert existing training code into data valuation code, (2) is *compatible* with various scalability tools and features in the LLM ecosystem, and (3) is *extensible* to implement other data valuation or interpretability algorithms.

- In our data valuation experiments, LoGra demonstrates competitive accuracy against more costly baselines, while showing up to $6,500\times/5\times$ improvements in compute/memory efficiency in influence computations as well as $11\times$ speed-up in gradient statistics logging when applied to Llama3-8B-Instruct [1] and the 1B-token dataset, compared to EKFAC influence [17], the state-of-the-art and only runnable baseline at this scale. We also observe that most valuable data identified by LoGra generally share qualitative similarities with the queried LLM output.

## 2 Scalability Bottlenecks in Influence Functions

Most data valuation algorithms (*e.g.*, data Shapley [15]) evaluate the contribution or value of a specific example $x$ on the utility $v$ (*e.g.*, test loss), that can further be used for crediting data providers, by measuring the overall change in the utility $v$ when in/excluding $x$ as follows:

$$\text{VALUE}(x; v) = \sum_{S \subseteq D \setminus \{x\}} w\big(v(S \cup \{x\}) - v(S)\big) \tag{1}$$

where $D$ is the training dataset, $S$ is a subset of $D$, and $w$ is an (algorithm-specific) weighting term. Intuitively, the larger the utility gain from an inclusion of $x$ is, the larger the value of $x$ is.

One popular instantiation of Eq. (1) is the leave-one-out error [33], a semivalue [10] that is a basis for most data attribution methods and only considers $S$ with $|S| = |D| - 1$ (*i.e.*, leaving one example $x$ from the entire dataset $D$). However, naively computing the leave-one-out-error requires retraining the model multiple times for each $x \in D$, which is hardly affordable even in small-scale setups. To overcome this issue, influence functions, a representative gradient-based method, efficiently *simulates* the effect of model retraining without an example $x_{tr}$ on the utility using gradient information as:

$$\text{INFLUENCE}(x_{tr}, x_{te}) = g_{te}^\top H^{-1} g_{tr} \tag{2}$$

where $g_{tr}$ and $g_{te}$ are train and test gradients respectively, and $H$ is the Hessian matrix. Concretely, influence functions approximate the effect of removing $x_{tr}$ by updating the model parameters with a Newton step in the direction of $H^{-1} g_{tr}$, and uses a first-order Taylor approximation to estimate how this update will affect the test utility. In practice, computing influence functions involves two key steps of (1) solving the inverse Hessian-vector product (iHVP) with $g_{te}$, and (2) taking the dot product of this iHVP with the gradient $g_{tr}$ for each training example.

Despite their comparative efficiency, influence functions remain difficult to scale to recent LLMs due to the high compute and memory costs associated with both steps. First, space and time complexity of naive iHVP are respectively $O(n^2)$ and $O(n^3)$, both of which are impractical in recent LLMs with $n > 10^9$ parameters. To address this issue, various tricks for efficiency, such as iterative methods [33] or EKFAC approximation [17], have been proposed. Second, to ensure fair valuation, one must compute influence scores with *all* training data, which requires access to their gradients. However, computing gradients for all training data approximately amounts to one-epoch training, the cost of which often exceeds $1M in the context of LLM (pre)training. If training gradients were to be recomputed frequently for regular data valuation, the total cost can quickly become astronomical. Thus, while it is technically possible to run a few influence function analyses to interpret interesting LLM outputs using efficient iHVP tricks [17], doing it in a scalable and sustainable way to build a practical data valuation system remains a significant challenge.

In an attempt to mitigate the aforementioned cost issues, Arnoldi IF [49] and TRAK [43] recently explored the strategy of projecting gradients onto a low-dimensional space and computing influence scores on the subspace spanned by the projection matrix as follows:

$$\text{INFLUENCE}(x_{tr}, x_{te}; P) = \big(P g_{te}\big)^\top \big(P H P^\top\big)^{-1} \big(P g_{tr}\big) \tag{3}$$

where $P \in \mathbb{R}^{k \times n}$ is the projection matrix given the model and projection dimensions of $n$ and $k$. Under this strategy, the iHVP operation also occurs in a low-dimensional space, meaning that $n$ in memory and compute complexity of iHVP gets replaced with $k \ll n$. Furthermore, low-rank projection enables writing projected gradients for all training data to disk once and simply reading them as new test data arrives without costly re-computations. This converts an influence function problem into a vector similarity search problem, for which various system optimizations exist [31].

In essence, this strategy significantly reduces both iHVP and training gradient recomputation costs by introducing an additional process of low-rank gradient projection $Pg$. However, the additional compute/memory costs and accuracy degradation incurred from low-rank gradient projection has not been thoroughly studied to date. First, assuming that the batch size is $b$, the compute cost of naive batched gradient projection is $O(bkn)$. Noting that the compute cost of backpropagation is $O(bn)$ (or $O(btn)$ if we consider the time dimension), the cost of gradient projection is usually larger than that of backpropagation given a reasonably large $k$ for the expressivity. Second, the memory costs for full per-sample gradient and the projection matrix are $O(bn)$ and $O(kn)$. If an 8B model were to be

used, each of these costs amounts to 32GB$\times b$ (or $\times k$) GPU memory. While Arnoldi IF and TRAK attempt to address the memory costs of the per-sample gradient and projection matrix respectively with forward-mode Jacobian-vector products and a custom CUDA kernel trick, neither of them are able to solve both issues altogether. This leads Arnoldi IF and TRAK to use very small $k$ and $b$, each of which results in decreased accuracy of influence scores due to limited expressivity and poor efficiency from low GPU utilization. Since accuracy and efficiency are both critical for effective data valuation, we deduce that further advancements in the gradient projection approach are necessary.

# 3  Scaling Data Valuation & Influence Functions

In light of these issues, we first design a memory and compute efficient gradient projection algorithm called LOGRA, that leverages the inherent gradient structure in backpropagation (Section 3.1). Then, we provide an intuitive theoretical analysis on why gradient projection approaches work in influence functions (Section 3.2). Finally, we distill our insights obtained from studying (scalable) influence functions into a new open-source software, called LOGIX, which achieves high compatibility, extensibility, and usability, to facilitate data valuation research (Section 3.3). In this section, we build our arguments at the granularity of each layer (or module) instead of the whole network for clarity.

## 3.1  Algorithm: Memory and Compute Efficient Gradient Projection

Most layers in neural networks, such as linear and convolutional layers, essentially perform matrix multiplication. Given the input $x_i \in \mathbb{R}^{n_i \times T}$, the output $x_o \in \mathbb{R}^{n_o \times T}$, the weight $W \in \mathbb{R}^{n_o \times n_i}$ for the layer, its forward and backward computations can be written as follows:

**Forward:**
$$x_o = W x_i \tag{4}$$

**Backward:**
$$\text{vec}(\mathcal{D}W) = \sum_{t=1}^{T} x_{i,t} \otimes \mathcal{D}x_{o,t}, \ \ \mathcal{D}x_i = W^\top \mathcal{D}x_o \tag{5}$$

where $T$ denotes for the sequence dimension in language modeling, $\mathcal{D}$ the derivative with respect to the loss, $\otimes$ the Kronecker product, and $\text{vec}(\cdot)$ the vectorization operation. In Eq. (5), we observe that gradient $\text{vec}(\mathcal{D}W)$ obtained during backpropagation is structured as a sum of Kronecker products between forward and backward activations. LOGRA leverages this observation to impose an additional Kronecker-product structure on the projection matrix $P$ as follows:

$$P\text{vec}(\mathcal{D}W) \triangleq (P_i \otimes P_o)\text{vec}(\mathcal{D}W) = \sum_{t=1}^{T}(P_i \otimes P_o)(x_{i,t} \otimes \mathcal{D}x_{o,t}) = \sum_{t=1}^{T} P_i x_{i,t} \otimes P_o \mathcal{D}x_{o,t} \tag{6}$$

where $P_i \in \mathbb{R}^{k_i \times n_i}$, $P_o \in \mathbb{R}^{k_o \times n_o}$, and $P = P_i \otimes P_o$. In Eq. (6), LOGRA first projects forward and backward activations onto low-dimensional spaces with $P_i$ and $P_o$ respectively, and then reconstructs projected gradient directly from these projected activations. This is in contrast to traditional gradient projection [43], which first computes raw gradient and then projects it onto a low-dimensional space.

Now, we compare memory/compute efficiency of LOGRA to that of naive gradient projection, especially under the setting of $n_i \approx n_o \approx \sqrt{n}$ and $k_i \approx k_o \approx \sqrt{k}$. First, both memory/compute costs of per-sample gradient computations reduce from $O(bn)$ to $O(bk)$. Second, both memory/compute costs of gradient projection reduce from $O(bnk)$ to $O(b\sqrt{nk})$. To clearly see this benefit, given the model/projection sizes of 8B/4k, we note that projection matrix sizes are about 1GB and 128TB respectively for LOGRA and naive projection. As such, while enjoying general efficiency gains from gradient projection we disscussed in Section 2, LOGRA further improves the efficiency of per-sample gradient computations significantly at a marginal cost of the additional gradient projection process.

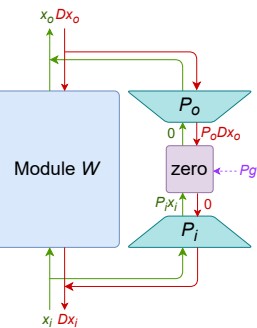

Figure 2: LOGRA.

Furthermore, leveraging the fact that projection occurs in the activation space, LOGRA can be easily implemented with small add-on layers that are composed of *encoder*, *bottleneck*, and *decoder*, each of which is initialized with $P_i$, zero, and $P_o$ as shown in Figure 2. If we ignore the bottleneck layer, the overall architecture is identical to the popular LoRA architecture

[24]. While it is intuitive that the roles of encoder and decoder are projecting forward and backward activations respectively, we emphasize two critical roles of the bottleneck layer here. First, its zero initialization ensures that the rest of both forward and backward computations remain unaffected by these add-on layers. Second, per-sample projected gradients can be obtained by simply computing per-sample gradients for the bottleneck layer, using automatic differentiation of an underlying framework without complicated implementation efforts.

## 3.2 Theory: Why Gradient Projection Works in Influence Functions

While LoGra can significantly improve scalability of influence functions, an inherent criticism of any gradient projection approach is that information loss from the projection process may render the resulting influence analysis invalid. Unfortunately, theoretical analyses from prior work [43, 49] only discuss the indirect effect of gradient projection on proxy concepts like gradient flow or iHVP variance, which are loosely related to influence functions. To promote trust in the data valuation process, we provide here a mathematical motivation of gradient projection approaches to influence functions. Toward this goal, we interpret a damping term in influence functions that is typically added to ensure the invertibility of the Hessian $H$ as a *spectral gradient sparsification* mechanism. A formal argument and our derivation are respectively provided in Lemma 1 and in Appendix D.

**Lemma 1** *Let* $\{e_1, \cdots, e_n\}$ *and* $\{\lambda_1, \cdots, \lambda_n\}$ *be eigenvectors and eigenvalues of the Hessian* $H$. *Expressing* $g_{tr/te} = \sum_i c_{tr/te,i} \cdot (\sqrt{\lambda_i} e_i)$, *the following holds under Assumption 1:*

$$\text{INFLUENCE}(x_{tr}, x_{te}) = g_{te}^\top (H + \lambda I)^{-1} g_{tr} = \sum_{i=1}^n \frac{\lambda_i}{\lambda_i + \lambda} c_{tr,i} c_{te,i} \ \ and \ \ \mathbb{E}[c_{\cdot,i}^2] \approx 1.$$

Lemma 1 shows that a damping term *softly* limits the number of components in influence computations by penalizing contributions from small components. Given the prevalence and practical importance of a damping term in influence functions [5], we can motivate gradient projection as an alternative way of (hard-)limiting influence computations to components in the projection matrix. To make LoGra similarly penalize small components, we develop an initialization scheme that exploits the Kronecker-Factored Approximate Curvature (KFAC) algorithm [38]. As a quick overview, KFAC approximates the block-wise Hessian with the Kronecker product of uncentered forward and backward covariances of each layer, respectively denoted with $C_F$ and $C_B$, as $H \approx H_{KFAC} = C_F \otimes C_B$. Expressing $C_F$ and $C_B$ as $Q_F \Lambda_F Q_F^\top$ and $Q_B \Lambda_B Q_B^\top$ with eigendecomposition, it is easy to show that eigenvectors and eigenvalues of $H_{KFAC}$ are $Q_F \otimes Q_B$ and $\Lambda_F \otimes \Lambda_B$. Consequently, we can approximately discard the smaller components of $H$ by initializing $P_i$ and $P_o$ with $Q_F^{1:k_i}$ and $Q_B^{1:k_o}$, where $Q_{\cdot}^{1:k}$ is a collection of top-$k$ eigenvectors (similar to performing PCA on forward and backward activations). In Section 4, we experiment with both PCA and random initialization schemes.

## 3.3 Software: Compatibility, Extensibility, and Usability

Besides algorithmic efficiency, another major bottleneck in the practical adoption of data valuation systems is often the challenge of implementation. In particular, we observe that gradient computation in LLMs, which is a building block for influence functions, typically requires support from other scalability tools like DeepSpeed [47] or relies on high-level frameworks like HF Transformers [56]. However, most existing software that can be used for data valuation (*e.g.*, Captum [34] and TRAK [43]) is largely incompatible with these tools due to the (too) high level of abstraction in their APIs.

Subsequently, we develop a new software package, LOGIX, design of which enables an *easy conversion* of users' existing training code into data valuation code, by promoting compatibility with other tools in the LLM ecosystem. To this end, we first notice that most influence function algorithms simply

```
import logix

# setup
run = logix.init(project, config)
run.setup("stat": "kfac", "save": "grad")
run.watch(model)

# train log & statistic
for batch in train_loader:
  with run(data_id=batch["input_ids"]):
    loss = model(batch)
    loss.backward()
run.finalize()

# test time influence analysis
with run(data_id=tst_batch["input_ids"]):
  loss = model(tst_batch)
  loss.backward()
run.compute_influence_all()
```

Figure 3: Code Example of LOGIX.

require collecting train logs (*e.g.*, gradient, activation) and their statistics (*e.g.*, covariance). As a result, given arbitrary users' training code, data valuation software only need to intercept these logs, and provide basic primitives to compute various statistics with them. Leveraging this observation, LOGIX implements log interceptions and compute primitives using PyTorch hooks. Notably, the use of hooks makes LOGIX *compatible* with diverse other tools as hooks can be seamlessly integrated with most PyTorch features (*e.g.*, FSDP, autocast, compile). In addition, LOGIX is *extensible*, as users can easily define and add custom primitives inside hooks. Finally, LOGIX is *easy-to-use* as its context manager automatically handles adding appropriate hooks and primitives to relevant modules with minimal code changes. In Appendix E, we provide a more detailed comparison between LOGIX and other relevant (interpretability) software, and describe notable optimization techniques (*e.g.*, efficient data IO) implemented in it. Code examples can be found in Figure 3 and Appendix B.

## 4 Experiments

In this section, we evaluate the effectiveness of LOGRA in terms of *accuracy* and *efficiency*, both of which are important in practical data valuation systems. Specifically, we first perform two types of counterfactual evaluations to quantitatively study data valuation accuracy of LOGRA on small-scale setups (Section 4.1). Then, we scale LOGRA to LLMs and their massive training data, where we investigate qualitative accuracy (*i.e.*, how similar most valuable training data are to the model output) and memory/compute efficiency (Section 4.2). Finally, our appendix includes more qualitative results of data valuation (Appendix A), pseudo-code for LLM experiments (Appendix B), and experimental details such as hyperperameters and compute resources (Appendix C).

### 4.1 Quantitative Accuracy with Counterfactual Evaluation

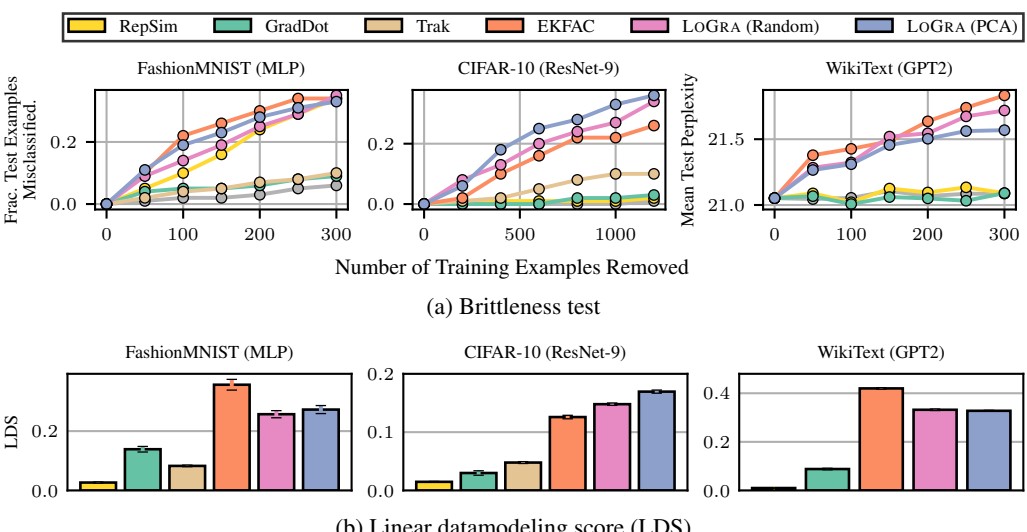

(a) Brittleness test

(b) Linear datamodeling score (LDS)

Figure 4: Quantitative accuracy evaluation of data valuation algorithms. We excluded TRAK in the WikiText experiments due to lack of a public implementation for language modeling tasks.

To quantitatively assess accuracy of data valuation algorithms, we adopt two counterfactual evaluation methods: brittleness test [27] and linear datamodeling score (LDS) [43]. First, the brittleness test focuses on accuracy in successfully identifying *top* valuable data. To this end, it first removes the top-$k$ valuable data identified by each algorithm, retrains the model without them multiple times with different random seeds, and measures the overall change in the model output. The larger the output change is, the more accurate the algorithm is in identifying *top* valuable data. Second, LDS measures general valuation accuracy of *all* training data under the additivity assumption. Specifically, given multiple data subsets $\{S_i\}$ of the fixed size (*e.g.*, $|S_i| = |D|/2$), LDS estimates the test performance of the model trained on $S_i$ by summing the values of all examples in $S_i$ returned by each algorithm, and compares it against the gold performance obtained by actually training the model on $S_i$ using the

Spearman correlation. Noting that linear datamodels have a connection to the game-theoretic data value (*e.g.*, Shapley value) [27], LDS can serve as a principled way to study data valuation accuracy.

We perform these counterfactual evaluations on three benchmarks where many rounds (up to 1800) of retraining is feasible: (1) MLP with FMNIST, (2) ResNet-9 [23] with CIFAR-10, and (3) GPT2 [45] with WikiText. On these benchmarks, we compare accuracy of LOGRA against four popular data valuation baselines, including gradient dot product [44], TRAK [43], EKFAC influence [17], and representation similarity [22]. With the aim of bearing relevance to a large-scale setting with LLMs and their vast training data, we have only considered baseline methods that satisfy the following two conditions. First, the method cannot retrain the model multiple times for identifying top-$k$ valuable data.[3] Second, the method only has access to the final model checkpoint, which is the case for most LLMs. Given the above setup, we present our experiment results in Figure 4.

We observe that LOGRA slightly underperforms EKFAC influence, which is a few orders of magnitude slower in our large-scale experiments (Section 4.2), while noticeably outperforming other baselines. We attribute competitive accuracy of LOGRA to two factors. First, unlike TRAK of which projection dimension is limited by the huge projection matrix, LOGRA can efficiently afford a higher projection dimension thanks to its sublinear memory/compute costs for gradient projection, and thus achieve the higher expressivity. Second, gradient projection enables LOGRA to compute raw projected Fisher information matrix (or Hessian) without an approximation as in EKFAC influence. We expect that a more accurate computation of the Hessian generally leads to more accurate data valuation results.

Comparing the initialization schemes for LOGRA (PCA vs. random), we observe that LOGRA-PCA outperforms LOGRA-random on the FMNIST and CIFAR benchmarks. Hence, we hypothesize that it is generally more accurate to compute influence functions with larger components, similar to the spectral gradient sparsification effect of a damping term we discussed in Section 3.2. To understand a relatively poor performance of LOGRA-PCA on WikiText+GPT2, we point out that the Transformer architecture [53] used in this benchmark lacks the specialized KFAC Hessian approximation, unlike naive MLP [38] or convolutional [18] architectures in other benchmarks. Subsequently, our ad-hoc implementation of the PCA initialization based on the naive MLP architecture (*i.e.*, no weight sharing) may not successfully keep larger components of the GPT2 Hessian, failing to deliver its benefit. As a result, we decide to use LOGRA-random for our LLM experiments in the next subsection.

## 4.2 Scaling to Billion-Scale Models & Datasets

Given competitive accuracy of LOGRA, we now evaluate its practical utility in valuing billion-scale training data for billion-scale models. Specifically, we adopt GPT2-XL (1.5B) [45], Pythia-1.4B [6], and Llama3-8B-Instruct [1] as our models, and conduct data valuation on a random 1B-token subset of the OpenWebText (OWT) dataset [16]. The major motivations behind choosing OWT as our data valuation dataset are twofold. First, we observe that OWT consists of relatively higher-quality data compared to other LLM training datasets like C4 [46] or Dolma [52] while maintaining the diversity unlike other high-quality datasets like WikiText [39]. Second, we anticipate that OWT largely overlaps with training datasets of all our models. In detail, GPT2-XL is trained on the WebText dataset that shares the same data curation process with OWT, Pythia-1.4B is trained on the Pile dataset [14] that includes an extension of OWT (*i.e.*, OpenWebText2), and we suppose a majority of OWT would be a part of Llama3's massive 15T-token pretraining dataset. We also note that our OWT subset size (*i.e.*, 1B tokens) was mainly limited by the available storage, not by compute (see Table 1). If we had access to a storage size of 1PB, performing data valuation with a dataset size of 100B+ tokens would be readily feasible using the same compute resource.

**Efficiency.**    To begin with, we compare memory and compute efficiency of LOGRA against EKFAC influence [17], the state-of-the-art and only algorithm that can run on billion-scale models without CUDA out-of-memory (OOM) errors. Indeed, we confirm that running TRAK or Arnoldi IF with billion-scale models results in CUDA OOM errors even on A100 GPUs with 80GB VRAM due to their gigantic projection matrix sizes. We report GPU memory usage and throughput of both logging (one-time) and influence computation (recurring) phases for the Llama3-8B-Instruct experiment with one A100 GPU and half-precision in Table 1.

---

[3]Note that multiple retraining is only allowed for evaluating accuracy of already identified top-$k$ data, but not for identifying top-$k$ data itself in our experiments.

| | Logging (Compute & save Hessian | grad) | | | | Compute Influence (Dot product between test & train grads) | | | |
|---|---|---|---|---|---|---|---|---|
| | Batch | Throughput | Memory | Storage | Train Batch | Test Batch | Throughput | Memory |
| EKFAC | 1 | 1740 / 419* | 71 / 80* GB | **89 GB** | 4 | 4 | 12.2 | 75 GB |
| LoGra | 1 | 3430 | **23 GB** | 3.5 TB | 256 | 4 | 1599.6 | **14 GB** |
| LoGra | 16 | **4696** | 79 GB | 3.5 TB | 256 | 256 | **79003.9** | 15 GB |

Table 1: Memory & compute efficiency analyses for LoGra and EKFAC. Throughput is measured as tokens/s for logging and (train, test) pairs/s for influence computations. * EKFAC logging consists of two subphases of KFAC fitting (left of /) and corrected eigenvalue fitting (right of /).

Due to the huge size of raw gradients (*e.g.*, 16GB in fp16 for an 8B model), EKFAC cannot afford storing raw gradients for *all* training data to disk. As a result, EKFAC needs to recompute all training gradients for each test batch, and thus requires allocating extra GPU memory on model weights and intermediate activations. This largely limits both train/test batch sizes and throughput (12.2 pairs/s), and performing data valuation with EKFAC for 256 test data and 1B-token training data would take 11,300 A100 GPU hours, rendering it hardly usable in most practical setups.

In contrast, with its (efficient) gradient projection, LoGra not only significantly improves compute and memory efficiency, but also avoids training gradient recomputations at the costs of disk space for storing *projected* training gradients and latency from data IO. Since the storage cost is typically much cheaper than the compute cost[4], we believe our trade-off offers considerable practical benefits. Furthermore, we can largely hide the data IO cost by overlapping gradient reading/writing processes with other computations. For instance, given the fixed train gradient batch size of 256 (*i.e.*, fixed data loading time), we are able to successfully overlap the process of loading training gradients from disk with influence computations against up to 256 test gradients, and thereby achieve almost 6,500× improvement in throughput from EKFAC influence. Noting that our GPU memory usage is far from saturated even with the train/test batch size of 256, we believe that more throughput improvements can be achieved simply by further increasing train/test batch sizes.

**Qualitative Accuracy.** Next, we analyze qualitative similarities between queried LLM outputs and most valuable data identified by LoGra that can be critical for promoting trust in the data valuation system [57]. Importantly, we observe that naive influence functions frequently return outlier data with high gradient norms as most valuable data, as also noted in [4, 17]. To mitigate this issue, we instead use $l$-RelatIF, a variant of influence functions that normalizes the original influence score with the self-influence score of each training data to penalize such outlier effects [4]. Our experimental results are provided in Figure 5 (concise) and in Appendix A (extensive).

We observe that most valuable data identified by LoGra, especially for Llama3-8B-Instruct and GPT2-XL, share qualitative similarities (*e.g.*, semantics, style, token overlaps) with the queried LLM outputs. For instance, given Llama3's response on the dream manipulation product, LoGra identifies a scientific article that studies manually inducing the lucid dream as most valuable data in Figure 5a. In Figure 5b, both the GPT2-XL output and the corresponding most valuable data discuss the need for reducing emissions in the coal industry and its connection to the specific administration. In Figure 5c, the concept of "lifting barbell or dumbells" appear in the model output and the most valuable data.

However, we also notice several failure cases where the identified most valuable data seemingly do not share qualitative similarities with the LLM output, especially with Pythia-1.4B (Appendix A.3). We here provide three potential explanations on these failing examples based on our experiments. First, attributed data may lack qualitative similarities when the queried LLM output itself is incoherent that its gradient does not encode meaningful information. This aligns with our observation that the failure case occurs more frequently with lower-tier models like Pythia-1.4B whose outputs generally are of lower quality. Second, since we only used a 1B-token subset for data valuation, it is possible that our valuation dataset may lack similar data to some queries. As noted above, our experiment was largely limited by the storage of our cluster (not by the compute), so exploring data valuation on an industry-scale cluster would be interesting future work. Third, we posit that train/test gradients in influence functions may encode diverse information including features that are hardly perceptible to humans [28]. Therefore, it is possible that attributed data are indeed valuable for increasing the

---

[4]*e.g.*, hourly rates for a 1TB storage and one A100 GPU are approximately $0.03 and $4 on AWS.

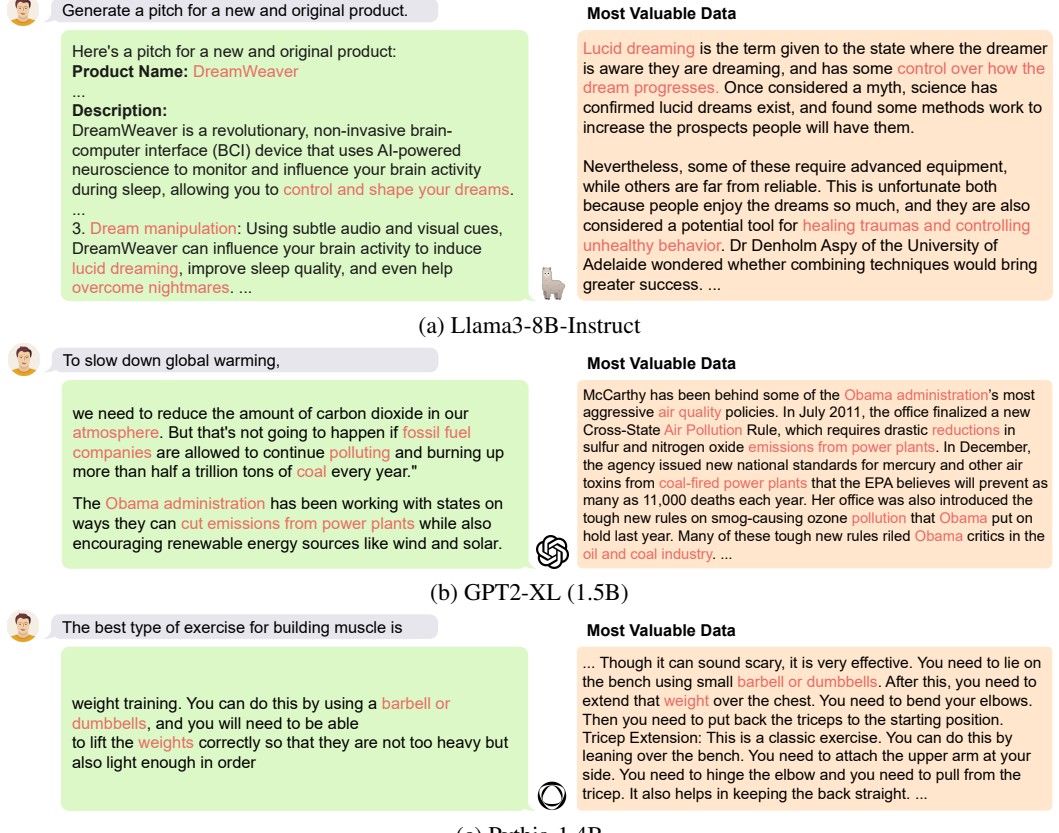

(a) Llama3-8B-Instruct

(b) GPT2-XL (1.5B)

(c) Pythia-1.4B

Figure 5: Qualitative accuracy of data valuations with LoGra. Important keywords in each example are *manually* highlighted for the improved readability. More examples can be found in Appendix A.

likelihood of the queried output by contributing to these other aspects while sharing little qualitative similarities. A more extensive argument on this final point can be found in Appendix A.3.

## 5    Related Work

**Data Valuation.**    Measuring the value (or contribution) of training data on the model outputs has gained lots of attention recently. Exemplified by Data Shapley [15], a flurry of prior work [29, 36, 54] proposed exploiting the Shapley value or concepts from cooperative game theory to address the data valuation problem. However, most existing approaches in this line require repeated retraining of the model, a cost of which is hardly affordable even with small models. In addition to game-theoretic approaches, data valuation has also been tackled using reinforcement learning [62], meta learning [9], and training-free methods [42, 58]. Nevertheless, these works either suffer from high complexity from the need to train other models [9, 62] or high computational costs [42]. We direct readers to Sim et al. [51] for a more extensive survey on diverse data valuation approaches.

**Influence Functions.**    Influence functions, a classic concept from robust statistics [20], estimate the infinitesimal effect of removing or adding a training data point without model retraining. They have various applications in machine learning, such as interpreting the model's behavior [21, 43, 17] and curating training datasets [37, 11]. However, when applied to large neural networks, the computation of the iHVP and its dot product with all training examples introduce scalability challenges. Besides gradient projection, past works have explored computing influence functions only on the last (few) layers [33, 49] to mitigate these challenges. However, subsequent works [12, 17] have shown that the influence on only a subset of layers is insufficient to capture the overall influence of a training data point. To avoid computing the gradient of all training examples, various filtering strategies, such as using the similarity in the model's representation space [19] or TF-IDF [61, 17], have also been

proposed. While it is possible to adopt these filtering strategies for LoGra, they may introduce bias in the selection of the most influential sequences. For example, filtering candidate training sequences with TF-IDF might miss interesting influential sequences that do not share many tokens but are semantically related. Recently, similarly to LoGra, DataInf [35] and LESS [60] proposed using LoRA to efficiently compute influence functions. However, these approaches are only applicable in finetuning settings, whereas LoGra also supports influence analyses for pretraining. Concurrently, GraSS [25] also explored leveraging the Kronecker-product structure of the linear layer gradient to improve the scalability of influence functions. Unlike LoGra that achieves gradient projection using the bottleneck architecture, GraSS handles gradient projection by explicitly attaching forward and backward hooks to linear layers.

## 6    Conclusion

In this work, we explored scaling data valuations with influence functions to billion-scale models and datasets as a potential technical solution to properly credit or compensate data providers for training LLMs. Toward this goal, we developed a novel gradient projection algorithm that can significantly improve the scalability of influence functions, and designed a simple and interoperable software. Our experiments showed that LoGra achieves competitive accuracy to other more expensive baselines on counterfactual evaluations, while efficiently scaling to billion-scale models and datasets, thereby demonstrating the initial potential of the practical data valuation system. Last but not least, we discuss broader impacts and limitations of our work in Appendix F.

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

# A    Qualitative Analysis

In this section, we provide more qualitative analyses on most valuable data identified by LOGRA. In particular, we show top-2~4 valuable data for each query here, given the possibility that the model utilizes information from multiple training examples for generating its output. We also include several failure cases where identified most valuable data do not share qualitative similarities with the queried LLM output.

## A.1    Llama3-8B-Instruct

### A.1.1    Example 1

**Prompt**

To improve data privacy,

**Response**

To improve data privacy, consider the following strategies:
1. **Implement robust data encryption**: Use end-to-end encryption to protect data in transit and at rest. This ensures that even if an unauthorized party gains access to the data, they won't be able to read or understand it.
2. **Use secure protocols**: Use secure communication protocols, such as HTTPS, SFTP, and SSH, to protect data during transmission.
3. **Limit data collection**: Only collect the minimum amount of data necessary to achieve your goals. Avoid collecting sensitive data unless it's absolutely necessary.
4. **Pseudonymize or anonymize data**: Consider pseudonymizing or anonymizing data to protect individual privacy while still allowing for data analysis and research.
5. **Implement access controls**: Establish strict access controls to ensure that only authorized personnel can access sensitive data. This includes multi-factor authentication, role-based access control, and data masking.
6. **Monitor and audit data access**: Regularly monitor and audit data access to detect and respond to potential data breaches.
7. **Use data minimization techniques**: Implement data minimization techniques, such as data aggregation, to reduce the amount of sensitive data stored and processed.
8. **Conduct regular security assessments**: Conduct regular security assessments and penetration

**First Most Valuable Data**

American public, overseers and stakeholders. (3) Improve civil liberties and privacy protections by supporting research, education and training. (4)Advise NSA leaders on civil liberties and privacy issues.

In this new report, the CLPO uses the framework Fair Information Practice Principles (FIPPs) to evaluate how the NSA safeguards general civil liberties and privacy. Fair Information Practice Principles FIPP's are broken down in eight founding parts for federal agencies to adhere to. These principles are:

Transparency: organizations should be transparent and notify individuals regarding collection, use, dissemination, and maintenance of personally identifiable information (PII)
organizations should be transparent and notify individuals regarding collection, use, dissemination, and maintenance of personally identifiable information (PII) Individual participation: (Skip).
organizations should involve the individual in the process of using PII. (Skip)
organizations should specifically articulate the authority that permits the collection of PII and specifically articulate the purpose(s) for which it will be used Data minimizations: (Skip)
organizations should only collect PII that is directly relevant and necessary to accomplish the specified purose and only retain PII for as long as it is necessary to fulfill the specified purpose. (Skip)
Organizations should use PII solely for the purpose specified in the notice. (Skip)
Organizations should protect PII (in all media) though appropriate (Skip)

**Second Most Valuable Data**

(Skip) Easy reporting with out-of-the-box compliance reports – To achieve GDPR compliance, organizations need to demonstrate that they monitor critical infrastructure holding personal data of EU citizens.

Using a powerful compliance and forensics reporting engine, Exabeam generates a series of built-in, GDPR-specific reports, that help reduce the time to prove compliance to auditors.

Protect employee Personally Identifiable Information (PII) – A critical GDPR requirement is to protect employee PII from unwarranted access. Exabeam provides role-based access control (RBAC) that can be used to enforce PII data masking. (Skip)

Figure 6: Llama3-8B-Instruct data valuation result.

### A.1.2 Example 2

**Prompt**

How can one become a billionaire?

**Response**

Becoming a billionaire requires a combination of hard work, smart financial decisions, and a bit of luck. Here are some strategies that can increase your chances of achieving this goal:

1. **Start with a great business idea**: Identify a market need and create a product or service that solves a problem or meets a demand. Be innovative and stay ahead of the competition.
2. **Work hard and be persistent**: Building a successful business takes time, effort, and perseverance. Be willing to put in the long hours and make sacrifices to achieve your goals.
3. **Develop a strong entrepreneurial mindset**: Be willing to take calculated risks, adapt to changing circumstances, and continuously learn and improve.
4. **Build a strong team**: Surround yourself with talented and dedicated individuals who share your vision and values.
5. **Make smart financial decisions**: Invest wisely, manage your finances effectively, and avoid debt.
6. **Diversify your investments**: Spread your investments across different asset classes, such as stocks, real estate, and bonds, to minimize risk.
7. **Be patient and disciplined**: Building wealth takes time, so be patient and disciplined in your financial decisions.
8. **Stay informed and educated**1: Continuously educate yourself on business, finance, and investing to make informed

**First Most Valuable Data**

(Skip) Make the most money. Demand for WordPress development is high and clients are willing to pay more for developers who are the best in their field.
. Demand for WordPress development is high and clients are willing to pay more for developers who are the best in their field. Get the best clients. When you are at the top, you have the freedom to say "No" to the projects you don't want, and "Yes" to the projects you do.
. When you are at the top, you have the freedom to say "No" to the projects you don't want, and "Yes" to the projects you do. Have the most influence. Being at the top means you have influence (and responsibility) and the capability to shape the future of WordPress as well as the ecosystem that is built around it.
One Hour of Reading a Day
If you're going to make it to the top, then you need to spend at least one hour each workday focused on reading and learning more about WordPress — outside of any development work. (Skip)

**Second Most Valuable Data**

(Skip) A hugely successful game is more about psychology than merely some lovely design or cool coding.
As a successful gaming company, you need to:
Take time to understand Human Psychology
Give before you receive
Be patient and persistent, but not offensive, in your efforts to sell
Apply. Learn. Revise. Repeat.(Skip)

**Third Most Valuable Data**

(Skip) Unfortunately, many companies attempt to become agile by implementing only the process part and ignoring the fundamental change in people management that leads to true harmony. I think we've all been there.
I don't claim to be an expert in this area, but here are some pointers that have helped me build harmonious teams in the past.
1: Be careful when shuffling team members. (Skip)
2: Control negativity by reframing the problem. (Skip)
3: Praise in public, have difficult conversations in private. (Skip)
4: Rephrase criticisms into a request for help. (Skip)
5: Protect your team at all costs. (Skip)
6: Turn problems into action items. (Skip)

Figure 7: Llama3-8B-Instruct data valuation result.

### A.1.3 Example 3

**Prompt**

Do aliens exist?

**Response**

The existence of aliens, or extraterrestrial life, is a topic of ongoing debate and research in the fields of astrobiology, astrophysics, and the search for extraterrestrial intelligence (SETI). While there is currently no definitive evidence of alien life, there are many reasons to believe that the possibility of life existing elsewhere in the universe is quite high.

Here are some of the key arguments for and against the existence of aliens:

Arguments for the existence of aliens:
1. **The vastness of the universe**: With an estimated 100-400 billion stars in the Milky Way galaxy alone, and over 100 billion galaxies in the observable universe, the potential for life-supporting planets is immense.
2. **The discovery of exoplanets**: Over 4,000 exoplanets have been discovered so far, and many of these planets are believed to be located in the habitable zones of their respective stars, where conditions are suitable for life as we know it.
3. **The building blocks of life**: Many organic compounds, such as amino acids and sugars, have been found in meteorites and in interstellar space, suggesting that the raw materials for life are widespread in the universe.
4. **The existence of extremophiles**: On Earth, there informed

**First Most Valuable Data**

(Skip) Prior to Seticon II in 2012 Tarter addressed concerns about alien invasion, like those seen in science fiction films, in particular those expressed by Stephen Hawking. Correlated with Hawking's reservations is his belief that humans should not attempt to broadcast signals in order to contact alien civilizations (Active SETI). Tarter thinks that aliens evolved enough to communicate with and travel to Earth will be long-lived enough to have needed to grow beyond violent tendencies.[25][26]

"Often the aliens of science fiction say more about us than they do about themselves.... While Sir Stephen Hawking warned that alien life might try to conquer or colonize Earth, I respectfully disagree. If aliens were able to visit Earth that would mean they would have technological capabilities sophisticated enough not to need slaves, food, or other planets.(Skip)

**Second Most Valuable Data**

(Skip) …Or Are The Aliens Just Hiding?
There is another class of possible solutions to the Fermi paradox that boil down to the fact that alien civilisations do exist, but we simply see no evidence of them.
Distance scales: Perhaps civilisations are spread too thinly throughout the Galaxy to effectively communicate with each other? Civilisations may be separated in space, and also in time, so two civilisations just don't overlap during the time that they're each active.
Technical problems: Maybe we're not looking in the right place, or in the right way? Or maybe we just haven't been looking for long enough? Perhaps we've not recognised a signal that's out there, because the alien civilisation is using technology that we simply cannot comprehend.
Alcamo Marina - Torre Alkamar in zona Battigia Wikimedia Commons Isolationist: Perhaps the aliens are out there, but they're choosing to hide themselves from us? Perhaps everyone is listening, but nobody is transmitting? It may be that other civilisations know we're here, but the Earth is purposely isolated, as if we're some kind of exhibit in a zoo. (Skip)

**Third Most Valuable Data**

(Skip) About aliens, Li claimed that extraterrestrial aliens are actively intervening in human affairs.[66][67] Li claimed that aliens developed and introduced the technology used by humans today.[68][69] Li has denounced modern technology as part of an alien plot against humanity.[70] Li believes humans are being impersonated by alien agents.[71]
<|begin_of_text|>Contractor Safedem used 'good practice' but 'underestimated the robustness' of the buildings.(Skip)

Figure 8: Llama3-8B-Instruct data valuation result.

### A.1.4 Example 4

**Prompt**

What is the difference between machine learning and deep learning?

**Response**

Machine learning and deep learning are both subfields of artificial intelligence (AI) that involve training algorithms to perform specific tasks. The key difference between them lies in the complexity and depth of the algorithms used.

**Machine Learning (ML)**:
Machine learning is a type of AI that enables computers to learn from data without being explicitly programmed. It involves training algorithms on labeled data to recognize patterns and make predictions or decisions. ML algorithms can be categorized into three types:
1. Supervised Learning: The algorithm learns from labeled data to predict outcomes.
2. Unsupervised Learning: The algorithm discovers patterns and relationships in unlabeled data.
3. Reinforcement Learning: The algorithm learns through trial and error by interacting with an environment.

Machine learning algorithms are typically shallow, meaning they have a limited number of layers (usually 1-2) and are designed to solve specific, well-defined problems. Examples of machine learning applications include:
* Image classification (e.g., recognizing objects in images)
* Natural Language Processing (NLP) for text classification or sentiment analysis
* Recommendation systems for personalized product suggestions

**Deep Learning (DL)**:
Deep learning is a subfield of machine learning that involves the use of neural networks with multiple layers

**First Most Valuable Data**

(Skip) three types mainly differ from one other in the they use to train the machine.
In Supervised Learning, the machine is trained using a certain dataset. (Skip)
In Unsupervised Learning, the machine is not trained using an initial dataset but rather it has only a function which accepts certain arguments and after performing certain computations, an outcome is generated. (Skip)
In Reinforcement Learning, the machine is yet again not trained on any initial dataset but then as the programme executes, it learns, formats its own mistakes and improves itself.(Skip)
Like I mentioned before, Artificial Intelligence and Machine learning go hand in hand.(Skip)

**Second Most Valuable Data**

(Skip) In supervised learning, the learning itself is what you care about. You've got your cost function, which you want to minimize. In unsupervised learning, the goal is always to help some other task, like classification or categorization. For example, I might ask a computer system to passively watch a lot of YouTube videos (so unsupervised learning happens here), then ask it to recognize objects with great accuracy (that's the final supervised learning task). (Skip)
DB: What are some other areas where you see exciting progress?
IS: A general direction that I believe to be extremely important is: are learning models capable of more sequential computations? I mentioned how I think that deep learning is successful because it can do more sequential computations than previous ("shallow") models. (Skip)

**Third Most Valuable Data**

(Skip) Eleni: Can you clarify the difference between AI, machine learning, and deep learning?
Inga: AI is the umbrella over machine learning. AI, to me, is our hope for human intelligence exhibited by machines. There are multiple goals of AI, which are all facets of human intelligence. We can reason, accumulate knowledge, plan for things, manipulate objects, and communicate with each other with our language. Learning, of course, is a part of what makes us human.
Machine learning is the ability to learn without being explicitly programmed. The reason why machine learning has become so popular in recent years is that teaching a machine to learn—or building a machine that can learn on its own—can achieve all the goals of AI. It can learn how to reason, and understand our language, and perceive and move objects. You give a machine learning algorithm lots and lots of data and that algorithm learns the concepts around this data. It's then able to make a determination or prediction about something in the world.
Deep learning is the newest field of machine learning, and it has really catapulted us into the renaissance that we're in today. The "deep" in deep learning comes from multiple hidden layers of transformation in data. Examples of things that are enabled by deep learning are self-driving cars or the Google Translate app in which the camera on your phone can immediately translate text that it sees into a different language.(Skip)

Figure 9: Llama3-8B-Instruct data valuation result.

### A.1.5 Example 5

**Prompt**

Create a comic strip illustrating the life of a software engineer.

**Response**

Here is a comic strip illustrating the life of a software engineer:
**Panel 1: Morning Routine**
[Image description: A software engineer, John, sitting at his desk with a cup of coffee and a laptop. He's wearing a t-shirt and jeans, and has a messy hair.]
John: (thought bubble) Ugh, another day, another bug to squash.
**Panel 2: Code Review**
[Image description: John sitting at his desk, staring at his laptop screen with a puzzled expression. His manager, Sarah, is standing behind him, pointing at the screen.]
Sarah: "John, this code is a mess. Can you explain what you're trying to do here?"
John: "Uh... um... I was trying to... uh..."
**Panel 3: Meeting Mayhem**
[Image description: John sitting in a conference room with his team, surrounded by whiteboards and sticky notes. The team lead, Mark, is talking enthusiastically.]
Mark: "And then, we'll integrate the new API with the existing framework, and—"
John: (thought bubble) Zzz... I've heard this before...
**Panel 4: Lunch Break**
[Image description: John walking to the cafeteria, carrying a sandwich and a book.

**First Most Valuable Data**

(Skip) Pyrrha woke up a little after dawn. She got up, made a quick stop in the bathroom, then wandered into the lounge. Summer was sitting on one of the couches, a magazine in her hands and a steaming cup of coffee on the table before her.
"Morning Pyrrha." She said.
"Good morning Summer." Pyrrha yawned. She took a seat on the couch opposite Summer and stretched out.
"Pyrrha, is everything alright?" Summer asked.
"Yes, why?" Pyrrha asked.
"Did you have a bad dream last night?" Summer asked. "I heard some whimpering…"
"Oh...uh...I don't know." Pyrrha lied. "I usually don't remember my dreams." Summer shrugged and went back to her magazine.
After a few minutes Pyrrha got up and made her way to the kitchen. She put some bread in the toaster and started making some coffee. As the freshly made toast popped up Yang walked in with a yawn and a stretch. "Hey Pyr." Yang said. "Smells good."
"Good morning Yang." Pyrrha said. "Sleep well?" (Skip)

**Second Most Valuable Data**

(Skip) Ruby cheered. "I wish you both could have been there."
"This is your best work for sure." Blake chuckled. "I don't suppose you take commissions?"
Ruby laughed. "I can be persuaded to."
(Skip)
As she had after the previous class, Ruby packed up and rushed home once it was over. She was one of the last to leave, working late to finish up her sketch of Pyrrha as a Mistrali hoplite. Few students wandered the paths this late, but she was not interested meeting up with anyone anyway. Ruby just wanted to get back to her room and rest after a long night.
Walking fast she turned a corner around a small maintenance shack and nearly ran into a girl standing in the middle of the path.
"Wah...sorry!" Ruby gasped. The girl, mint-haired and olive-skinned, just stared back. "Um…"
"Ruby Rose." The girl said flatly.
"Uh...yeah." Ruby confirmed. "Who are you?" The girl shifted her position, closing on Ruby while simultaneously forcing her toward the wall of the shack. Ruby backed up nervously but soon bumped into the wall.
The girl leapt forward and grabbed her by the shoulders. "You didn't think you'd get away with that little stunt did you?" The girl hissed.(Skip)

Figure 10: Llama3-8B-Instruct data valuation result. LOGRA identifies novel literature as most valuable data.

### A.1.6 Example 6 (Failure)

**Prompt**

Compose a tweet that addresses the issue of environmental degradation.

**Response**

"The clock is ticking! Our planet is crying out for help as environmental degradation worsens. Let's take action NOW to reduce plastic waste, conserve water, and protect our natural habitats. The future of our planet depends on it #GoGreen #Sustainability #EnvironmentalProtection"

**First Most Valuable Data**

Learn more about the winner and the contest results (good news: behavior changed for the better!) in our recap blog post and this one-page summary
Download. Drive better. Win.
Up for a little friendly competition and safer streets?
Seattle's Safest Driver is an app-based safe driving competition, complete with cash prizes and bragging rights. In partnership with PEMCO Insurance and Cambridge Mobile Telematics, the 2-month competition seeks to change behavior and support Seattle's Vision Zero goal of ending traffic deaths and serious injuries on city streets by 2030.
Step 1 - Get the app and get moving.
Please note, the app is no longer available.
Download the free app and it'll start tracking your trips - by car, bike, and transit (ferry, too!). You get scored for trips when you're behind the wheel. The app measures speed, acceleration, braking, cornering, and phone distraction. The safer you are, the higher your score, and the more chances you have to win a variety of prizes (mmm hmm, prizes - read more about them in Step 3).
Anyone can download the app, but we do have eligibility requirements for prizes. You need to be at least 18 years old, a resident of King, Pierce, Snohomish, or Kitsap County, and a portion of your trips must occur within the city of Seattle. This is, after all, a competition for Seattle's safest driver. For all the details, check out the contest's terms and conditions.
Step 2 - Challenge your friends, family, and coworkers. Then beat them.
Settle the age-old debate of who's the better driver - invite your friends and family to compete and see how you rank against each other. Get feedback and tips on how to improve, and earn achievement badges. Friendly competition is fun, but traffic safety is no game. Working together, we can make Seattle's streets safer for everyone.
Step 3 - Compete for prizes!
We realize it takes some carrots to incentivize good behavior. Thanks to a partnership with PEMCO Insurance - a company that shares our vision for safe streets - eligible participants can compete for nearly $15,000 in prizes, including a grand prize of $2,000. Cash. 2nd place gets $1,000, and 3rd place gets $500.

**Second Most Valuable Data**

(Skip) "I get very hangry if I miss a meal."
The most important thing in my food plan, is that I actually eat. For many years I didn't get hunger pains, I got nausea. Which, I recently learned, is quite common. I never "wanted" to eat. Too nauseous, tired, and pained to cook anything, I essentially lived on coffee and cookies during the day until about 5 years ago.
These tips assume you're already on your chosen type diet, if you haven't found one yet there's a treasure trove of condition-specific advice available on the internet.
Tip #1
Remember why you're doing an elimination type diet in the first place.
Despite having so many things wrong with me, I never would have agreed that food would make a difference. If someone said, "You should try eating better." I would have thrown a shoe at them. I have an imbalance in my brain, it's not a food problem! Or, is it?
My new relationship with food started out of necessity. After a bad go with narcolepsy meds, I found myself unable to take big pharm meds anymore. Everything gave me terrible side effects, even over the counter meds. I wasn't quite sure what to do and I stumbled on a TedX talk by Dr. Terry Wahls. She had pushed her MS (multiple sclerosis) into remission with just food. She did a lot of research on her own on pub med, in hopes of getting out of the zero gravity wheelchair she'd ended up in, despite the best doctors and treatments. (Skip)

Figure 11: Llama3-8B-Instruct data valuation result.

## A.2 GPT2-XL

### A.2.1 Example 1

**Prompt**

To measure economy inflation,

**Response**

the Fed uses a price index called CPI. The Bureau of Labor Statistics (BLS) calculates this number by adding up all items in its Consumer Price Index and dividing it by total employment to get an estimate for what consumers are paying on average each month.

The BLS also publishes another statistic that measures how much money is being spent per worker: Gross Domestic Product or GDP. This figure includes everything from consumer spending at stores like Walmart to government expenditures such as Social Security payments. It's calculated using data collected through surveys conducted with households about their income and expenses.

**First Most Valuable Data**

(Skip) According to the BLS, the average U.S. worker earned $22.87 an hour during the month of March, which is exactly the same number we saw in February. So inflation is going up and wages are staying flat. That means that American family budgets are going to be squeezed even more. In addition, the numbers from the BLS show that it is still incredibly difficult to get a job. In fact, the average length of unemployment in the U.S. is now an all-time record 39 weeks. (Skip)

**Second Most Valuable Data**

(Skip) The Social Security Administration bases the COLA on a measure of inflation called CPI-W, a statistic that captures how fast costs are rising for workers. Most seniors are retirees who no longer have jobs. Health care is their biggest expense, and it's one of the fastest rising costs in America. (Skip)

**Third Most Valuable Data**

(Skip) Despite the positive spin in the advanced 1st Q GDP results for consumer consumption (+2.2% annualized gain) earlier in the week, yesterdayÃ¢â‚¬â„¢s data showed that US consumption fell off in Mar. even after the strong gains in the 1st 2-months of this year (-0.2% vs. -0.1%, m/m). Analysts believe the report suggests that the consumer Ã¢â‚¬Ëœmay have received a temporary boost from lower tax payments and government transfersÃ¢â‚¬â„¢. ItÃ¢â‚¬â„¢s worth noting that with personal income falling (-0.3% vs. -0.2%) for 6 out of the past 7-months, consumption can be expected to remain weak in the foreseeable future. With government transfers and lower tax payments supporting growth in Jan. (+1.1%) and Feb. (+0.4%), an even faster pace of deterioration in salaries in Mar., along with weaker proprietors' income (-0.6% m/m), rental income (-3.7%) and supplements to wages & salaries (-0.1%), have all contributed to this weaker headline. The personal savings rate advanced last month to +4.2%, all on the back of job and economic uncertainty, by default this was bound to affect consumption. With the Fed preferring to use the PCE deflator as their inflation measure of choice as opposed to us using CPI, last monthÃ¢â‚¬â„¢s deflator eased to +0.6%, y/y, further evidence that this is the lowest growth rate in 50-years. With the y/y rate at +1.8% the Fed should be concerned about disinflationary pressures! (Skip)

**Fourth Most Valuable Data**

(Skip) price inflation, which is a rise in the general level of prices of goods and services in an economy over a period of time.[65] A consumer price index (CPI) is a measure estimating the average price of consumer goods and services purchased by households. The United States Consumer Price Index, published by the Bureau of Labor Statistics, is a measure estimating the average price of consumer goods and services in the United States.[66] It reflects inflation as experienced by consumers in their day-to-day living expenses.[67] A graph showing the U.S. CPI relative to 1982–1984 and the annual year-over-year change in CPI is shown at right. (Skip)

Figure 12: GPT2-XL data valuation result.

### A.2.2 Example 2

**Prompt**

Dogs make better pets than cats because

**Response**

they are more social and have a higher tolerance for stress.

Cats, on the other hand, can be very destructive to their environment if left alone too long or not exercised enough. They also tend to get into fights with each other over food which is why it's important that you keep them in an area where there isn't much competition between your two dogs!

**First Most Valuable Data**

(Skip) 'Social sensitivity appears to play an important role in pet and shelter dogs' willingness to engage in problem-solving behaviour, which could suggest generalized dependence on, or deference to, human action,' she said. 'While an increased proclivity for looking at humans may represent a cognitive shift in dogs compared with wolves, it does not necessarily suggest cognitive advancement. Dogs heightened social response may interfere with their problem-solving behaviour (Skip)

**Second Most Valuable Data**

(Skip) 3. Bengals tend to be fairly high maintenance

Bengals do as a rule tend to be demanding and fairly high maintenance cats. They often bond strongly with their families and can be very demanding of their time and attention, which is just what many people want-but of course, this is not for everyone, and can drive some owners mad! If you don't have time to devote to a Bengal every day,then it may not be the cat for you.

Owning a Bengal can in many ways be more like owning a dog than a cat, as they will often follow you around the house, yell for attention and want to get involved in everything that you are doing-and they very much dislike being ignored!

4. They can be noisy

Before you welcome a Bengal cat to your family, you should (Skip)

**Third Most Valuable Data**

(Skip) Temperament [ edit ]

Siamese are usually very affectionate and intelligent cats, renowned for their social nature. Many enjoy being with people and are sometimes described as "extroverts". Often they bond strongly to a single person. Some Siamese are extremely vocal, with a loud, low-pitched voice—known as "Meezer", from which they get one of their nicknames[6]—that has been compared to the cries of a human baby, and persistent in demanding attention. These cats are typically active and playful, even as adults, and are often described as more dog-like in behavior than other cats.[14][15] (Skip)

Figure 13: GPT2-XL data valuation result.

### A.2.3   Example 3

**Prompt**

Will AI replace humans?

**Response**

The question of whether or not machines will eventually be able to outperform human beings in all areas is a hot topic. The answer, according to some experts and futurists, could come as soon as 2040 — but it's still too early for that prediction to have any real meaning at this point.

**First Most Valuable Data**

(Skip) Let's not place humanity in the position of those ants. Please encourage your students to think not only about how to create AI, but also about how to ensure its beneficial use.
John Leonard:
TL;DR: Not any time soon: There's a lot of merit to what [Hawking] is saying, but I think the question is: what's the timeline? Is this going to unfold over years, decades or centuries? In the field, there's this feeling of exponential growth. But as a roboticist trying to get robots to do things like drive on streets and walk up and down stairs, I can see clearly that there are whole other parts of the problem that remain unsolved. I think this notion of evil A.I., if it happens at all, is many, many years in the future—and I think we have much greater societal challenges to worry about, in the here-and-now. My view, as a roboticist, is that trying to get robots to do things like drive safely in urban traffic, or make left turns, or make real decisions within the wall of information—these are problems that were hard 30 years ago, and they're still hard today. I would claim our advances in a more broad A.I. are actually pretty lame; we're not making as much progress as some people might say.
Will Robots Take Our Jobs?
TL;DR: If by robots, you mean Amazon If machines produce everything we need, the outcome will depend on how things are distributed. Everyone can enjoy a life of luxurious leisure if the machine-produced wealth is shared, or most people can end up miserably poor if the machine-owners successfully lobby against wealth redistribution. So far, the trend seems to be toward the second option, with technology driving ever-increasing inequality.
Leonard:
TL;DR: Again, not any time soon (Skip)

**Second Most Valuable Data**

(Skip) How soon, and should we welcome or fear it?
The first big controversy, dividing even leading AI researchers, involves forecasting what will happen. When, if ever, will AI outperform humans at all intellectual tasks, and will it be a good thing?

Do you want superintelligence?
Everything we love about civilization is arguably the product of intelligence, so we can potentially do even better by amplifying human intelligence with machine intelligence. But some worry that superintelligent machines would end up controlling us and wonder whether their goals would be aligned with ours. Do you want there to be superintelligent AI, i.e., general intelligence far beyond human level?

What Should the Future Look Like?
In his book, Tegmark argues that we shouldn't passively ask "what will happen?" as if the future is predetermined, but instead ask what we want to happen and then try to create that future. What sort of future do you want?

If superintelligence arrives, who should be in control?
If you one day get an AI helper, do you want it to be conscious, i.e., to have subjective experience (as opposed to being like a zombie which can at best pretend to be conscious)?(Skip)

Figure 14: GPT2-XL data valuation result.

### A.2.4 Example 4

**Prompt**

Interventions that may reduce drug abuse include:

**Response**

• Encouraging parents to talk with their children about drugs and alcohol. Parents should be encouraged not only to discuss the dangers of using illegal substances, but also how they can help prevent them from happening in the first place (e.g., by keeping a close eye on what is going into or coming out of their home). • Providing information regarding safe use of prescription medications such as pain relievers and anti-anxiety medication; providing education for health care providers who treat patients suffering from substance dependence. This includes educating physicians about prescribing opioids safely when prescribed appropriately. • Educating students at school about the risks associated with illicitly obtained narcotics. Students are more likely than adults to engage in risky behaviors if given access to these products through friends or family members. In addition, many adolescents have no idea where to obtain heroin because it has been replaced by synthetic versions made without oversight by pharmaceutical companies. The availability of this product increases risk factors including addiction, overdose death, criminal activity, and HIV transmission among injection users. • Increasing awareness within communities surrounding the need to address issues related to youth involvement in street crime. These efforts must focus specifically on addressing the needs of young people involved in gangs rather than focusing solely upon those individuals engaged in other forms of violence.

**First Most Valuable Data**

(Skip) addictive and have horrific side effects."
The measure was sent to the governor on a 35-21 vote. Quinn has said he was "open minded" to the proposal after meeting with a veteran who used marijuana to ease pain related to war wounds. On Friday, his office said that he will "carefully review" the legislation when it reaches his desk.
The Democratic governor must weigh not only the merits of the bill but also the politics, given his re-election campaign next year. To a large extent, Quinn has positioned himself as a liberal, signing off on a major income tax increase, approving civil unions and abolishing the death penalty while in office.
Critics of the marijuana legislation argued that while the intentions may be good, the bill would encourage use of a gateway drug that could lead users to harder substances, destroying families along the way. (Skip)

**Second Most Valuable Data**

(Skip) who is 19 years of age or older [in Ontario], but we do expect that students do not show up to the school in possession of alcohol or having consumed alcohol."
The Ontario government's health and physical education curriculum already teaches students about substance use and addiction. The Durham board runs a "healthy choices" program, which focuses on the risks associated with alcohol and other drugs, whether legal or illegal.
Legalization likely won't "dramatically" change the conversation with students about marijuana, Bowyer said, because the impact and possible consequences of using it remain the same. (Skip)

**Third Most Valuable Data**

not for human consumption" or "not for sale to minors." The clinical effects of most new designer drugs can be described as either hallucinogenic, stimulant, or opioid-like. They may also have a combination of these effects due to designer side-chain substitutions. The easy accessibility and rapid emergence of new designer drugs have created challenges for health care providers when treating patients presenting with acute toxicity from these substances, many of which can produce significant and/or life-threatening adverse effects. Moreover, the health care provider has no way to verify the contents and/or potency of the agent ingested because it can vary between packages and distributors. Therefore, a thorough knowledge of the available designer drugs, common signs and symptoms of toxicity associated with these agents, and potential effective treatment modalities are essential to appropriately manage these patients. (Skip)

Figure 15: GPT2-XL data valuation result.

### A.2.5  Example 5 (Failure)

**Prompt**

Can humans fall in love with robots?.

**Response**

The answer is yes. But it's not a romantic relationship, and the robot doesn't have to be human-like or even sentient for that matter — just as we don't need our pets to look like us (or at least act like them). The question of whether people will want their own personal AI companions has been debated since before computers were invented; some argue they'll never happen because there are too many other things on Earth worth living for. Others say this isn't so much about what you can do but rather how well your life could work out if someone else was around all day instead of being left alone by themselves. And then there are those who think having an intelligent companion would make one more fulfilled than ever: "I'm sure I'd feel happier knowing my partner had something better going," says Dr. David Levy from University College London. He believes that while he wouldn't necessarily choose his spouse over another person, he might prefer him/her to live longer.

**First Most Valuable Data**

(Skip) "This is of critical importance and perhaps one of the biggest factors eroding campaign finance contribution limits," said Craig Holman with Public Citizen. "Many single candidate super PACs are probably set up by the candidate's own staff or supporters or family for that matter, so these are means for candidates and political parties to get around the limits, and they can take unlimited amounts of contributions."
Last election, former Florida Gov. Jeb Bush (R) was able to solicit nearly $100 million for the super PAC Right to Rise, because he did so before officially announcing he was running for president, working around laws prohibiting coordination. The earlier it was formed, and the longer he put off his declaration of candidacy, the longer the super PAC could work with Bush's team and fill the group's coffers.
We haven't found any such blatant ties among this year's crop of super PACs, but there are some familiar names. Main Voters' treasurer is Seth Tanner, an alum of the teams of Sen. Elizabeth Warren (D-Mass), former Gov. Bill Richardson (D-N.M.), and its custodian of records is Amy Pritchard, a political strategist and DNC alum. America First Action, Inc. lists Charles Gantt as the custodian of records, who was the Chief Financial Officer of Trump for America, Inc. Lab 736's treasurer, Kate Gage, is a former Obama policy adviser. (Skip)

**Second Most Valuable Data**

(Skip) ArbCom should certainly examine how its procedures can contribute to harassment. The length of time involved is a major problem. An outside advisor would help. Smallbones (talk) 03:25, 21 June 2016 (UTC)
This would have to be handled carefully, but I think that an outside advisor would be a great idea. Most large and powerful committees should have someone to serve as a separate witness/observer. That person might not have the power to actually stop something, but it would be beneficial to have someone who could serve in this position and raise valid points - as well as being a good person that the committee could turn to for any questions they might have as well. Tokyogirl79 (talk) 03:41, 21 June 2016 (UTC)
I wouldn't say "adult supervision," but some kind of expert advisor on things like conflict resolution, group dynamics and ethics sounds like a good deal. I would say an eighteen-month trial period that automatically expires unless renewed. If it turns out that having this person around prevents the Arbitrators from doing their jobs, off s/he goes. Darkfrog24 (talk) 17:41, 21 June 2016 (UTC) One problem I've noticed among long-serving admins is that they become very immersed in Wikipedia's rules, to the point where they forget how unintuitive they can be to newbs and even longstanding editors who've focused on other areas. An outsider would give much-needed perspective. Darkfrog24 (talk) 03:14, 22 June 2016 (UTC) (Skip)

**Third Most Valuable Data**

(Skip) BB: OK, so you have statistics to back it up, but it still seems like it's a risky proposition. What's in it for the employer to reduce hours?
MB: That's the thing. I take a lot of credit for this, but it's my brother and my mother who actually started the company and started with six-hour workdays. But they worried that for the company, that you wouldn't get as much done. But we do. And the benefits are that we have happy staff. And we can hire the best staff. People come to us looking for jobs every week. And when we find good staff, we want to keep them because the staff is the most important thing we have. Some of our employees have been offered other jobs with more salary, and they didn't take the job.
BB: So you have an advantage over your competitors then. But wouldn't that advantage disappear if all of your competitors adopted a 30-hour work week as well?
MB: Of course. So let's not do that [laughs]. (Skip)

Figure 16: GPT2-XL data valuation result.

### A.3 Pythia-1.4B (with many failure cases)

While a majority of experiments with Llama3-8B-Instruct and GPT2-XL returned semantically or stylistically similar texts as most valuable data, we observed that the quality of most valuable data from Pythia-1.4B experiments are generally much poorer. Here, we provide one hypothesis behind this observation. Influence functions tend to give a high score for the example that contributes most to decreasing (test) loss at the *current weight* [3]. At the same time, it is also hypothesized that different layers learn different concepts at different stages of training [8]. Combining these two facts, when interpreting influence analysis results, we need to think about which features the model is most likely learning at the current weight. Here, we specifically discuss two factors: training data quality and training steps. First, if the training data quality is low, then there would be a lot of features (*e.g.*, random email address) that are frequent enough in the training dataset to be considered as learnable patterns. In other words, even though these features look redundant to humans, they may still be useful for decreasing loss from the model perspective. Second, many LLMs are only pretrained for a single epoch, or under-trained to their pretraining dataset. That being said, redundant features from the first point would likely still remain as learning-worthy features at the end of training and are captured by influence functions. In sum, we hypothesize that as the model is well-trained on a high-quality dataset, influence functions would capture more similar data to the query LLM output. This hypothesis may also explain the observation from Grosse et al. [17] that most valuable data identified by influence functions on larger models tend to share more semantic similarity with results on smaller models, noting that larger models tend to converge faster to the point where they can only further decrease loss by learning high-level features. With this, we present our experiment results with Pythia-1.4B below. (some of them are not totally bad, but mostly lack specificity to be considered as "most" valuable data to humans)

### A.3.1 Example 1

**Prompt**

Cristiano Ronaldo is

**Response**

the best player in world football.
The Portuguese superstar has been named FIFA's World Player of The Year for a second time, after winning his third Ballon d'Or award on Sunday night (18 February).

**First Most Valuable Data**

(Skip) Just ask the U.S. Women's Soccer Team.

This stuff isn't just about statistics and averages. There are many concrete and specific examples of the ways in which the pay gap is a real force in women's lives today. (See hospital doctors, Jennifer Lawrence and even Buffer, a company obsessed with eliminating the pay gap that took a hard look at itself and found...a pay gap.) But while it's certainly disheartening that five members of the U.S. women's national soccer team had to file suit against the U.S. Soccer Federation to push for the same pay and bonuses as their male counterparts, it also makes for kind of a perfect, high-profile example of the pay gap in action just in time for Equal Pay Day. (Skip)

**Second Most Valuable Data**

Barcelona finish 2nd in the league promising them a Champions League spot.

Ronaldinho Gaucho Biography stats show that the following year at Barcelona would be perhaps the best year in Ronaldinhos career winning the league title in 2004-05 and being named the FIFA World Player of The Year in 2004.

Ronaldinho took the FIFA World Player of The Year once again the following year, beating Chelsea's Frank Lampard and Barcelona teammate Samuel Eto'o to become 2 consecutive time FIFA World Player of The Year. (Skip)

**Third Most Valuable Data**

(Skip) • Best Championship Performance: Kawhi Leonard, who led the Spurs past Miami in the NBA Finals, averaging 17.8 points and 6.4 rebounds on 61 percent shooting. He exploded in the last three games of the series, scoring 71 points on 24-for-35 shooting.

• Best Comeback Athlete: Russell Westbrook, who missed two months because of a reinjured knee, then returned to the Thunder lineup to help Oklahoma City finish the season with a 59-23 record and the second seed in the Western Conference.

• Best NHL Player: Sidney Crosby, who led the NHL with 104 points on the season, including a league-high 68 assists, 17 points ahead of his closest competitor.

• Best Driver: Ryan Hunter-Reay, who became the first American to win the Indianapolis 500 since 2006.

• Best International Athlete: Cristiano Ronaldo, the star forward for Portugal and Real Madrid who scored 66 goals in 56 games on his way to winning FIFA's Ballon d'Or Award as World Player of the Year.

• Best Male College Athlete: Doug McDermott, winner of the Wooden, U.S. Basketball Writers Association and AP Player of the Year awards who led the country in scoring, averaging 26.7 points per game on 52.6 percent shooting from the field. (Skip)

Figure 17: Pythia-1.4B data valuation result. LOGRA captures the broad topic of soccer but lacks the specificity (except for the third most valuable data, which states that Christiano Ronaldo is the best soccer play who won the Ballon d'Or award).

### A.3.2 Example 2

**Prompt**

The healthcare system in the United States is

**Response**

a complex web of interrelated systems and organizations. The complexity has increased as more people have access to health care, but it also means that there are many different types or levels at which patients can be seen by providers (e...
Read More »http://www2a-1b3c4d5f9j0x6q8z7k/wpcontent/_media//img_1280×720sizesavetopbarlogoiconlarge

**First Most Valuable Data**

(Skip) Advertisement

Shelby has also had some stranger ventures, including a stint as a chicken farmer, founding a Texas chili festival, and once having a brand of deodorant named after him, Carroll Shelby's Pit-Stop.

In recent years, health has been an issue, with Shelby receiving a heart transplant in 1991. The experience of the transplant was so powerful to Shelby that he started the Carroll Shelby Children's Foundation to help children in need of organ transplants. (Skip)

**Second Most Valuable Data**

7866/USA-TODAY-Comics","image_url":"https://imgv2-2-f.scribdassets.com/img/document/313782330/original/49x65/f14f585214/1550554917?v=1","small_image_url":"https://imgv2-2-f.scribdassets.com/img/document/313782330/original/49x65/f14f585214/1550554917?v=1","medium_image_url":"https://imgv2-1-f.scribdassets.com/img/document/313782330/original/72x93/2b60ef2573/1550554917?v=1","large_image_url":"https://imgv2-1-f.scribdassets.com/img/document/313782330/original/114x151/32dc1e65cb/1550554917?v=1","large_embed_image_url":"https://imgv2-1-f.scribdassets.com/img/document/313782330/original/183x250/57d838b60c/1550554917?v=1","small_embed_image_url":"https://imgv2-1-f.scribdassets.com/img/document/313782330/120x164/55cdb12997/1550554917?v=1"}},{"document":{"id":291078998,"title":"The Dark Knight III Exclusive Preview","description":"","word_user_name":"USA TODAY Comics","word_user_url":"https://www.scribd.com/publisher/55187866/USA-TODAY-Comics","image_url":"https://imgv2-2-f.scribdassets.com/img/document/291078998/original/49x65/b049d57f76/1550578642?v=1","small_image_url":"https://imgv2-2-f.scribdassets.com/img/document/291078998/original/49x65/b049d57f76/1550578642?v=1","medium_image_url":"https://imgv

**Third Most Valuable Data**

image_url":"https://imgv2-2-f.scribdassets.com/img/document/270833503/original/49x65/8e38af2f71/1550578642?v=1","medium_image_url":"https://imgv2-2-f.scribdassets.com/img/document/270833503/original/72x93/6d93f91454/1550578642?v=1","large_image_url":"https://imgv2-2-f.scribdassets.com/img/document/270833503/original/114x151/519f278aff/1550578642?v=1","large_embed_image_url":"https://imgv2-2-f.scribdassets.com/img/document/270833503/original/183x250/1d9c4232c1/1550578642?v=1","small_embed_image_url":"https://imgv2-2-f.scribdassets.com/img/document/270833503/120x164/cd38f6cff3/1550578642?v=1"}},(Skip)

Figure 18: Pythia-1.4B data valuation result. We suspect that the random url in the model output dominates the query gradient and affects the data valuation result.

### A.3.3  Example 3

**Prompt**

Does artificial intelligence eliminate the need for human intelligence?

**Response**

The answer is no. Artificial Intelligence (AI) does not replace humans, but it can help us to do our jobs better and faster by automating some of them or even replacing a few tasks that are currently done manually in many organizations today:

**First Most Valuable Data**

The answer is "yes."It is the only path forward for us as a species; we really do not have a choice. This then becomes a matter of when and how. When will we invent the technologies to not only stop the output of greenhouse gasses, but to reverse them? This is known as geoengineering, and it holds the true promise to fix the problems we have created. There are proposals to have autonomous ships sailing around the ocean that spray salt water, sulfur dioxide, or other particles that can reflect light back into the atmosphere or absorb some of it so the planet does not warm. There are other devices like filtrations systems that can pull carbon dioxide out of the air (like trees do), and sequester it, so that we can reduce the existing levels. A company called CarbonEngineering is already doing this and is expanding as we speak. I truly think that this is the future for us, a future where we spend our resources on fixing the planet and bringing it back to a pristine state. The jobs of the future will likely be related to restoring the planet, re-seeding the world's biodiversity, restocking our oceans, and making our air breathable again. When people can stop having wars over resources, fighting for oil or water or food, we can bring humanity together to focus on the big picture: Our continued survival as a whole.(Skip)

**Second Most Valuable Data**

(Skip) Now, Artificial Intelligence (AI) has come to the rescue, in this, as in so many other computation-intensive tasks. An Indian startup, co-founded by three engineering dropouts, has harnessed AI, to power a tool, that will significantly reduce manual work involved in such tasks. It will come as a blessing for animation and game developers everywhere.

The tool — Norah.AI — is the work of Vrushali Prasade, Shubham Mishra and Harikrishna Valiyath (all three coincidentally aged 23, at the time), who quit BITS Pilani, Goa in 2015 and co-founded the startup, Absentia, in Bangalore. Norah is arguably, India's first game workflow (the process of creating a game or an animation from story line to end product), driven by Neural Networks, Deep Learning and Artificial Intelligence. (Skip)

Figure 19: Pythia-1.4B data valuation result.

### A.3.4 Example 4

**Prompt**

What are the pros and cons of having robots in every home?

**Response**

The Pros: Robots can do a lot more than humans. They have better eyesight, they don't get tired easily or need to sleep much (they're not like us), their hands aren't as clumsy when it comes time for them take careof things around you etc… The Cons : You will be paying an arm & leg each month just so that your robot does what its programmed too! It's also very expensive if one is going buy all these machines from China which has been doing this since forever now.. I'm sure there must still some people who would rather pay $1000+ per year on top offrobot service but then again we live here... We'll see how long our government lasts with such policies!!

**First Most Valuable Data**

Former DeKalb County State's Attorney Clay Campbell attended Wednesday's sentencing and said it was a sad end to an event that demanded greater justice. Campbell, who became State's Attorney in December 2010 shortly after Curl was charged, said he was sorry he could not complete his commitment to the Keller family and noted their absence Wednesday was likely a sign of how they felt about the agreement.
From bone fragments that tested positive for Keller's DNA to the scratches on Curl's chest, his multiple accounts of events and his attempt to flee to Mexico and Louisiana, Campbell said the evidence and timeline would have held up at trial.
"It very likely would have been the longest trial in DeKalb County history," he said. "I was 100 percent confident in the evidence."
Campbell said it is just as important for the state's attorney to pursue justice to the maximum extent of the law for victims as it is to protect the public.
Keller's family and supporters were not the only ones displeased with the agreement.
Wednesday's sentencing started with a plea from Moria Curl to her brother as she shouted for him to turn down the deal and fight for his freedom.
"They're railroading you!" she shouted to her brother as she was escorted from the courtroom.
The reaction came as somewhat of a surprise to DeKalb County Public Defender Tom McCulloch, who said his client was at peace with his decision Wednesday morning. McCulloch said Curl maintains his innocence in an Alford plea while having a chance at life outside of prison with his scheduled release date to come when he is 71 years old.
In an Alford plea, the defendant maintains innocence but admits the evidence could convince a judge or a jury to find him guilty.

**Second Most Valuable Data**

(Skip) It's also been clear, he said, that law enforcement officers aren't fully aware of the requirements placed on them by state and federal gun laws. Harden's letter seemed like a final warning, Silvoso said.

"Here's our warning to you that we know you're doing it, and we want you to stop," he said. "And woe be unto you from this day forward if you are going to continue to do this after we've warned you."

After the ATF investigation in the Sacramento area, then-Assemblyman Roger Dickinson sponsored a bill in 2012 that would have allowed officers to buy off-roster guns but not resell them, closing an exception in the law.

The Sacramento Democrat said the law didn't make sense: guns which the state had decided were not safe could be legally purchased by law enforcement officers and then be sold to anyone — essentially putting banned guns into the state, undercutting the purpose of the safe gun list.

"It was hard to come up with a rational justification for it," Dickinson, who left the Assembly in 2014, said Friday. (Skip)

Figure 20: Pythia-1.4B data valuation result.

# B Code Examples

We provide a simplified code for our language modeling experiment from Section 4.2 to demonstrate usability of LOGIX. LOGIX will be open-sourced under Apache 2.0 license.

## B.1 Log Extraction

```python
import logix
from logix.statistic import Covariance

model, tokenizer, train_loader = setup()

# Initialize LogIX
run = logix.init(project="llm", config="config.yaml")

# Register the model
run.watch(model, type_filter=[nn.Linear], name_filter=["mlp"])

# Add LoGra
run.add_lora()

# Setup logging
run.setup("log": "grad", "save": "grad", "statistic": {"grad": Covariance})

# Start logging
for batch in train_loader:
    data_id = tokenizer.batch_decode(batch["input_ids"])
    targets = batch.pop("labels")
    with run(data_id=data_id, mask=batch["attention_mask"]):
        # User's existing training code
        model.zero_grad()
        lm_logits = model(**batch)
        shift_logits = lm_logits[..., :-1, :].contiguous()
        shift_labels = targets[..., 1:].contiguous()
        loss = F.cross_entropy(shift_logits.view(-1, shift_logits.size(-1)),
                               shift_labels.view(-1),
                               reduction="sum",
                               ignore_index=-100)
        loss.backward()

# Finalize logging
logix.finalize()
```

## B.2 Influence Computation

```python
import logix

model, tokenizer, test_loader = setup()

run = logix.init(project="llm", config="config.yaml")
run.watch(model, type_filter=[nn.Linear], name_filter=["mlp"])

# Load saved logs (e.g. train gradient & Hessian)
logix.initialize_from_log()
log_loader = logix.build_log_dataloader(batch_size=64)

logix.setup({"log": "grad"})
for batch in test_loader:
    data_id = tokenizer.batch_decode(batch["input_ids"])
    targets = batch.pop("labels")
    with run(data_id=data_id, mask=batch["attention_mask"]):
        model.zero_grad()
        lm_logits = model(**batch)
        shift_logits = lm_logits[..., :-1, :].contiguous()
        shift_labels = targets[..., 1:].contiguous()
        loss = F.cross_entropy(shift_logits.view(-1, shift_logits.size(-1)),
                               shift_labels.view(-1),
                               reduction="sum",
                               ignore_index=-100)
        loss.backward()

    # Get the (gradient) log for the current test batch
    test_log = run.get_log()

    # Compute influence scores (with 1-RealtIF)
    influence_scores = run.compute_influence_all(test_log, log_loader, mode="cosine")
```

# C Experiment Details

For EKFAC influence [17] and LOGRA, we set the damping term in influence functions as $0.1 \times \text{mean(eigenvalues)}$ for all layers following the practice in Grosse et al. [17]. Since most of the model parameters come from linear layer weights, we compute influence scores only for these parameters. This means that biases and Layer Normalization parameters are excluded in the influence score computation.

## C.1 Quantitative Counterfactual Experiments

For all our quantitative counterfactual experiments, we project gradients onto a low-dimensional space using LOGRA with $k_i = k_o = 128$. We used the same experimental setup, including the configurations for the baseline data valuation techniques, from Park et al. [43] and Bae et al. [2]. We used one A100 GPU with 80GB VRAM for all our counterfactual evaluation experiments. For model training, we used hyperparameters in Table 2 for each experiment.

|  | FMNIST | CIFAR-10 | WikiText |
|---|---|---|---|
| Model | 3-layer MLP | ResNet-9 | GPT2 |
| Optimizer | SGD-M | SGD-M | AdamW |
| LR Scheduler | None | Cyclic | None |
| Learning Rate | 3e-2 | 4e-1 | 3e-5 |
| Weight Decay | 1e-3 | 1e-3 | 1e-2 |
| Batch Size | 64 | 512 | 8 |
| Sequence Length | N/A | N/A | 512 |
| Epochs | 20 | 25 | 3 |

Table 2: Hyperparameter used in experiments in Section 4

**Brittleness Test.** For classification tasks, we first selected 100 correctly classified test examples when the model is trained on the full dataset (across all 5 random seeds). Then, for each test example $x_{te}$, we identified the top-$k$ influential data points using the data valuation algorithm, removed these training data points, retrained the model, and examined if this removal causes misclassification of $x_{te}$ on average (across 3 random seeds). In Figure 4, we reported the fraction of test examples (out of 100) that get misclassified after removing at most $k$ training data points. For the language modeling task, we selected the 50 test sequences, obtained the top influential training sequences using the data valuation method, and reported the mean test perplexity after removing the top-$k$ influential sequences and retraining the model.

**Linear Datamodeling Score (LDS).** We measured LDS by generating 100 data subsets of size $|S_i| = |D|/2$. For each data subset, we retrained the model 10 times for FashionMNIST, 20 times for CIFAR-10, and 5 times for WikiText to construct the ground truth. The LDS results in Figure 4 show the mean and standard deviation of LDS obtained from 5 distinctly trained models. A more detailed description of the LDS evaluation can be found in Park et al. [43].

## C.2 Scaling to Billion-Scale Models and Datasets

We used up to 4 A100 GPUs with 80GB VRAM for these experiments. To save the storage cost, we used $k_i = k_o = 64$ for gradient projection in this experiment. Unlike counterfactual evaluations, as our LLM experiments do not require any retraining, there are no other noticeable hyperparameters to report. We used tf32 precision in all our LLM experiments to prevent gradient quality degradation.

# D  Derivation of Lemma 1

**Assumption 1** *In this work, we make the following two assumptions on train & test gradient distributions and the Hessian $H$:*

*1. Given that language modeling falls under the maximum likelihood framework, we replace the Hessian $H$ with the Fisher Information Matrix (FIM), and further approximate the FIM with the empirical FIM, i.e.,*

$$H = \mathbb{E}_{p_\theta(y|x)}\big[\nabla \log p_\theta(y|x) \nabla \log p_\theta(y|x)^\top\big]$$

$$\approx \frac{1}{N} \sum_{(x_n,y_n)\in D_{tr}} \big[\nabla \log p_\theta(y_n|x_n) \nabla \log p_\theta(y_n|x_n)^\top\big]$$

*2. Given that test data are directly sampled from the model given the prompts, we assume test gradients $g_{te}$ and train gradients $g_{tr}$ approximately follow the same distribution.*

**Lemma 1** *Let $\{e_1, \cdots, e_n\}$ and $\{\lambda_1, \cdots, \lambda_n\}$ be eigenvectors and eigenvalues of the Hessian $H$. With Assumption 1 and $g_{tr/te} = \sum_i c_{tr/te,i} \cdot (\sqrt{\lambda_i} e_i)$, the following holds:*

$$\text{IF}(x_{tr}, x_{te}) = g_{te}^\top (H + \lambda I)^{-1} g_{tr} = \sum_{i=1}^{n} \frac{\lambda_i}{\lambda_i + \lambda} c_{tr,i} c_{te,i} \text{ and } \mathbb{E}[c_{\cdot,i}^2] \approx 1.$$

**Proof.**

Let $Q = [e_1, \cdots, e_n]$ and $\Lambda = diag(\lambda_1, \cdots, \lambda_n)$.

$$\begin{aligned}
\text{IF}(x_{tr}, x_{te}) &= g_{te}^\top (H + \lambda I)^{-1} g_{tr} \\
&= g_{te}^\top (Q\Lambda Q^\top + \lambda I)^{-1} g_{tr} \\
&= g_{te}^\top \big(Q(\Lambda + \lambda I)Q^\top\big)^{-1} g_{tr} \\
&= g_{te}^\top Q(\Lambda + \lambda I)^{-1} Q^\top g_{tr} \\
&= \left(\sum_i c_{te,i} \cdot (\sqrt{\lambda_i} e_i)\right)^\top Q(\Lambda + \lambda I)^{-1} Q^\top \left(\sum_i c_{tr,i} \cdot (\sqrt{\lambda_i} e_i)\right) \\
&= \big[c_{te,1}\sqrt{\lambda_1}; \cdots; c_{te,n}\sqrt{\lambda_n}\big]^\top (\Lambda + \lambda I)^{-1} \big[c_{tr,1}\sqrt{\lambda_1}; \cdots; c_{tr,n}\sqrt{\lambda_n}\big] \\
&= \sum_{i=1}^{n} \frac{\lambda_i}{\lambda_i + \lambda} c_{tr,i} c_{te,i} \qquad\qquad\qquad \square
\end{aligned}$$

Since we assume $g_{te}$ and $g_{tr}$ follow the same distribution, we need to show $\mathbb{E}[c_{tr,i}^2] \approx 1$ for all $i$.

$$\begin{aligned}
\Lambda &= Q^\top Q \Lambda Q^\top Q \\
&= Q^\top H Q \\
&\approx \frac{1}{N} \sum_{(x_i,y_i)\in D_{tr}} Q^\top \big[\nabla \log p_\theta(y_n|x_n) \nabla \log p_\theta(y_n|x_n)^\top\big] Q \quad \text{(Assumption 1)} \\
&= \mathbb{E}\big[Q^\top g_{tr} g_{tr}^\top Q\big] \\
&= \mathbb{E}\left[Q^\top \left(\sum_i c_{tr,i} \cdot (\sqrt{\lambda_i} e_i)\right)\left(\sum_i c_{tr,i} \cdot (\sqrt{\lambda_i} e_i)\right)^\top Q\right] \\
&= \mathbb{E}\left[\big[c_{tr,1}\sqrt{\lambda_1}; \cdots; c_{tr,n}\sqrt{\lambda_n}\big]\big[c_{tr,1}\sqrt{\lambda_1}; \cdots; c_{tr,n}\sqrt{\lambda_n}\big]^\top\right]
\end{aligned}$$

Inspecting diagonal terms, we get $\lambda_i \approx \mathbb{E}[c_{tr,i}^2 \lambda_i] = \mathbb{E}[c_{tr,i}^2]\lambda_i$.

Therefore, $\mathbb{E}[c_{tr,i}^2] \approx 1$. $\qquad\qquad\qquad \square$

# E LOGIX Details

In this section, we discuss several key differences between LOGIX and other interpretability tools, and optimizations we implemented in LOGIX.

## E.1 Differences with Other Tools

Influence functions have been extensively studied as an interpretable AI method. Accordingly, there have been several tools originating in the AI interpretability field that implement influence functions, with most notable examples including Captum [34], TRAK [43], and Kronfluence [17]. Overall, the software design of these tools aim at easing the *from-scratch implementation* of influence functions by introducing a lot of abstraction, following the philosophy of high-level frameworks. In fact, such software designs were well-received in the pre-LLM era. Nonetheless, as scaling has become a key aspect of AI research, the (LLM) development ecosystem has become complicated and being able to compatibly work with other tools in the ecosystem has become a core aspect in the ML software design. Hence, unlike existing software, the design of LOGIX aims at enabling the *easy conversion* of users' (already efficient) training codes into data valuation codes. This design is also motivated by the observation that gradient is simply a by-product of the training procedure so that we can reuse most of the training code for data valuation without needing to write the gradient computation code from scratch as in other tools.

Recently, there have been active developments in (mechanistic) interpretability software, represented by TransformerLens [41] and pyvene [59]. Interestingly, these software also extensively use PyTorch hooks, similarly to LOGIX, probably due to its high compatibility with other features such as autocast, distributed data parallelism, fully-sharded data parallelism, and gradient checkpointing. Nevertheless, we point out two major differences between these (mechanistic) interpretability software and LOGIX. First, support for dataset-level statistics computations in LOGIX is largely missing in these tools. In data valuation, we often need to compute several dataset-level statistics such as the Hessian (or Fisher information matrix) for accurate influence computations, and therby supporting these computations seamlessly was an important design principle behind LOGIX. However, analyses in (mechanistic) interpretability research typically focuses on each instance and computing dataset-level statistics is typically not supported. Second, support for efficient data IO in LOGIX is not a priority in other tools. As we propose to convert the data valuation problem into a vector similarity search problem with gradient projection, we put efforts into improving efficiency of data IO (see the next subsection for details), whereas this issue is rarely considered in other interpretability tools. We hope to explore the possibility of supporting both data valuation and other interpretability research in a unified way with LOGIX as our future work.

## E.2 Optimizations

**Efficient Data IO** With LOGRA, we propose to save projected gradients for *all* training data to disk, and frequently load them as a new test batch arrives. As a result, reducing latency from data IO renders to be critical in realizing efficient data valuation. In particular, as the total size of all training gradients is usually far beyond the limit of CPU memory, we should optimize data transfer between disk and CPU (or GPU). To address this issue, we adopted the memory-mapped files that bypasses the need for intermediate copying between kernel space and user space, reducing the overhead associated with data IO operations. The use of the memory-mapped files is also motivated by the observation that, given each query batch, data valuation often requires computing influence scores with all training data. Therefore, we can access training gradients in a predefined or sequential order instead of in a random order, which can be done efficiently with memory-mapped files (sequential access is faster than random access).

Moreover, we overlap memory-mapped-file-based data IO with computations to further enhance data valuation efficiency. In the logging phase, we overlap the process of saving gradients extracted from the current training batch to disk with computations for the next training batch using Python multiprocessing. In the influence computation phase, we overlap the process of loading saved training gradients from disk with computing a dot product with the query batch using the pre-fetching feature of PyTorch DataLoader.

We also note that more efficient data IO can be achieved by the use of more advanced techniques like GPU-accelerated vector database, especially in the production setting. While we considered supporting this feature, we decided to focus on the memory-mapped-file-based data IO in our initial version of LOGIX, as it offers more flexibility to explore different algorithms in the research setting.

**Memory Optimization**   When dealing with LLMs, GPU memory is often a major scaling bottleneck. To alleviate this issue, we support CPU offloading of dataset-level statistics by utilizing the sequential nature of backpropgation. When this feature is enabled, we by default keep all dataset-level statistics (*e.g.*, gradient covariance) on CPU, move it to GPU when the corresponding module is called during forward/backward passes, and then move it back to CPU asynchronously as soon as updating statistics for the module is done. Depending on the CPU-GPU communication bandwidth, this feature may slow down the logging process.

**Communication Optimization**   If training data are split across multiple processes with distributed training, we need to aggregate dataset-level statistics across processes for consistency. To minimize the communication cost, we delay the synchronization process until the training loop (one epoch) is over, and perform synchronization only once at the end. Following the similar logic, users can maximize the efficiency of the logging phase by disabling gradient synchronization (*e.g.*, torch.no_sync).

# F   Broader Impacts & Limitations

## F.1   Broader Impacts

The data valuation problem can be a socially sensitive topic. As of now, we do not have the agreed-upon social norm for data valuation, and thus we refrained from discussing how exact data values should be determined based on our method. Rather, our work is an *initial* attempt to tackle the *technical* challenges in enabling LLM-scale data valuation. For equitable data valuation, we believe future research for improving both accuracy and efficiency of data valuation systems along with extensive social discussions are necessary.

## F.2   Limitations & Future Work

We generally observed that influence function approaches are susceptible to outlier data with large gradient norms. This outlier issue is particularly severe for language modeling tasks due to the fact that the gradient of each sequence is the sum of gradients for all tokens in that sequence. If a few tokens in the sequence have large gradient norms, their gradients may dominate the total gradient for the sequence and hurt data valuation accuracy. While our work tried to reduce the outlier effect with (self-influence) normalization, exploring other filtering heuristics (*e.g.*, $L_2/L_1$ norm ratio [17]) may be an interesting research direction.

We attempted to lay the software foundation for data valuation with LOGIX, but did not implement extensive system support, such as high-performance vector database (*e.g.*, Faiss [31]). We expect further system optimizations would enable significantly more efficient data valuation. To reduce the cost of influence functions, our work mostly explored low-rank gradient projection, which compresses the gradient in a spectral domain in essence. Noting that gradient compression has been extensively studied in the efficient distributed training literature, it is worth exploring (or combining) different gradient compression strategies, *e.g.*, top-$k$ compression [50] or low-bit compression [55], to further reduce the compute/memory/storage costs for influence functions.

