# OpenReview forum: "What is Your Data Worth to GPT? LLM-Scale Data Valuation with Influence Functions"
_NeurIPS.cc/2025/Conference — NeurIPS 2025 poster_

### Official Review · Reviewer_ffNn · 2025-06-29

**Clarity:** 2
**Significance:** 3
**Originality:** 4
**Rating:** 4
**Confidence:** 4

**Summary:**

This paper proposes a data valuation framework for large-scale language models by introducing **LoGra**, a low-rank gradient projection method that significantly reduces the computational and memory overhead of influence functions. Alongside, the authors present **Logix**, a practical software toolkit that simplifies the integration of data valuation into existing training pipelines. Compared to existing methods, LoGra achieves orders-of-magnitude improvements in efficiency without sacrificing accuracy.

**Questions:**

The experiments are limited to models like Llama3-8B-Instruct, GPT2-XL, and Pythia-1.4B. Is it possible to evaluate the approach on larger-scale models (e.g., 30B or 65B) to further validate scalability?

The analysis of failure cases on Pythia-1.4B is insightful. However, could the authors propose concrete remedies or practical strategies to mitigate such issues in future applications?

Is the proposed acceleration method hardware-agnostic? For example, can similar improvements be expected on TPUs or different GPU architectures?

If the authors are able to address the above issues, I would be willing to raise my score.

**Ethical Concerns:**

["NO or VERY MINOR ethics concerns only"]

**Final Justification:**

Thank the author for resolving my confusion. This work has a good starting point and relatively significant contributions. However, there are still issues such as the small size of the test dataset and model. Based on the above opinions, the final score is given as follows.

**Limitations:**

yes

**Quality:**

3

**Strengths And Weaknesses:**

Strengths:

The paper provides numerous concrete examples to demonstrate the effectiveness of the proposed method, making the results more convincing.

It introduces a practical toolkit (Logix) to lower the barrier of adoption for data valuation in LLMs.

Additionally, the authors analyze failure cases, offering insightful guidance for future improvements.

Weaknesses:

The dataset used for large-scale experiments is limited both in variety and size, as only a 1B-token subset of OpenWebText (OWT) is tested. This may affect the generalizability of the results, especially to broader domains such as multilingual or multimodal datasets.

---

> ### Author Rebuttal · Authors · 2025-07-30
>
> Thanks for your reviewing effort! We attempt to address questions raised by the reviewer below:
>
> ---
>
> > Q1.1 The experiments are limited to models like Llama3-8B-Instruct, GPT2-XL, and Pythia-1.4B. Is it possible to evaluate the approach on larger-scale models (e.g., 30B or 65B) to further validate scalability?
>
> > Q1.2 The dataset used for large-scale experiments is limited both in variety and size, as only a 1B-token subset of OpenWebText (OWT) is tested. This may affect the generalizability of the results, especially to broader domains such as multilingual or multimodal datasets.
>
> A. While we successfully tested gradient statistics extraction and computation on models up to 13B parameters, we encountered substantial challenges when attempting experiments with larger datasets and models due to computational and storage constraints in our research environment—extracting and computing gradients requires resources comparable to actual model training. Given our focus on LLM pretraining data valuation, we conducted the most extensive experiments our resources permitted. Since computational budget can be approximated as constant x model_size x dataset_size, further scaling either dimension would have required compromising the other, making it practically infeasible within our constraints. We hope these resource limitations do not detract from the evaluation of our contributions.
>
> ---
>
> > Q2. The analysis of failure cases on Pythia-1.4B is insightful. However, could the authors propose concrete remedies or practical strategies to mitigate such issues in future applications?
>
> A. In Appendix F.2, we identified outlier tokens as a primary source of failure cases—these tokens produce gradients that dominate the entire sequence gradient and consequently skew influence scores. We propose several mitigation strategies including gradient normalization, clipping techniques, and filtering heuristics to address these issues. Generally, we believe developing influence computation strategies that are robust to outlier tokens would be a promising future research topic.
>
> ---
>
> > Q3. Is the proposed acceleration method hardware-agnostic? For example, can similar improvements be expected on TPUs or different GPU architectures?
>
> A. Yes, we believe our method is largely hardware-agnostic — neither LoGra nor the software design principles underlying LogIX contain any hardware-specific dependencies that would prevent compatibility with TPUs or alternative GPU architectures. Our experiments already demonstrate this flexibility to some extent, as we successfully deployed our approach across various GPU types including RTX 2080Ti, 3090, L40, A6000, and A100.
>
> Thanks again and please let us know if you have any other questions.

---

### Official Review · Reviewer_fJkS · 2025-06-30

**Clarity:** 3
**Significance:** 2
**Originality:** 3
**Rating:** 4
**Confidence:** 4

**Summary:**

This paper proposes a low-rank gradient projection algorithm named LOGRA, aiming to extend influence functions to LLM-scale data valuation tasks without significantly compromising accuracy. The authors also design LOGIX, a modular framework that seamlessly integrates data valuation procedures into standard training pipelines. Experiments demonstrate that this method substantially reduces computational and memory overhead while maintaining reasonable estimation quality.

**Questions:**

1.LOGRA's performance depends on the initialization of the Kronecker projection matrices (PCA or random). Is the method inherently less effective for architectures with less structured activations (e.g., Transformers with multi-head attention)?

2.Does LOGIX support more complex training pipelines, such as multi-stage pretraining followed by SFT or RLHF?

3.How do you assess whether the “valuable data” identified by LOGRA truly affect model behavior? Have you considered introducing metrics such as embedding coverage or goal-oriented perturbation?

**Ethical Concerns:**

["NO or VERY MINOR ethics concerns only"]

**Final Justification:**

Most concerns are resolved.

**Limitations:**

yes

**Paper Formatting Concerns:**

No major formatting issues were found. The paper follows NeurIPS 2025 formatting guidelines. All figures, tables, and footnotes appear properly placed and typeset. No action needed.

**Quality:**

2

**Strengths And Weaknesses:**

*Strengths*

1. Practical problem formulation: Data valuation is a highly relevant issue in the context of LLMs, especially with regard to legal attribution and compensation. The paper addresses the core challenge of extending influence function methods to modern large-scale models.

2. Complete engineering implementation: The LOGIX framework is carefully designed with compatibility in mind, supporting modern PyTorch ecosystems such as FSDP. This adds practical value to the work.

3. Well-executed quantitative experiments: The authors provide solid empirical evidence through standard metrics such as LDS and brittleness tests on small models.

*Weaknesses*

1. Limited generalizability: The effectiveness of LOGRA appears to rely on structural assumptions about the Hessian. For instance, PCA initialization performs poorly in Transformer architectures without weight sharing, raising concerns about broader applicability.

2. Subjective qualitative analysis: The identification of “valuable data” relies heavily on human inspection of semantic similarity between training samples and model outputs. A more systematic and quantitative evaluation metric is lacking.

3. Lack of ethical discussion: Despite the centrality of data ownership and provenance, the paper lacks a thorough discussion on potential misuse, legal concerns, or fairness implications of the proposed system.

---

> ### Author Rebuttal · Authors · 2025-07-30
>
> Thanks for your reviewing effort. We attempt to address questions raised by the reviewer here.
>
> ---
>
> > Q1/W1.LOGRA's performance depends on the initialization of the Kronecker projection matrices (PCA or random). Is the method inherently less effective for architectures with less structured activations (e.g., Transformers with multi-head attention)?
>
> A.Our results in Section 4.1 demonstrate that LDS scores for WikiText+GPT2 (using both random and PCA projections) actually exceed those obtained for FashionMNIST+MLP and CIFAR10+ResNet-9 experiments. This suggests that LoGra's effectiveness for architectures with less structured activations may not be inherently limited as the question implies. That said, we recognize that understanding the factors affecting data valuation accuracy—including the impact of Kronecker projection matrix initialization—represents a valuable direction for future research.
>
> ---
>
> > Q2. Does LOGIX support more complex training pipelines, such as multi-stage pretraining followed by SFT or RLHF?
>
> A. Yes, LogIX supports diverse training pipelines. Thanks to its modular software design, for instance, LogIX integrates seamlessly with Huggingface Trainer, which is commonly used for various post-training stages including SFT and other fine-tuning approaches.
>
> ---
>
> > Q3/W2. How do you assess whether the “valuable data” identified by LOGRA truly affect model behavior? Have you considered introducing metrics such as embedding coverage or goal-oriented perturbation?
>
> A. We extensively evaluated how valuable data identified by LoGra affects model behavior through our brittleness and LDS experiments in Section 4.1. Specifically, our brittleness experiments, which is recognized as one of strengths of our work by the reviewer, demonstrate that removing the most valuable data selected by LoGra results in non-trivial increases in loss values for target text examples. I assume this is conceptually close to the goal-oriented perturbation suggested by the reviewer (please correct us if we are wrong). If our writing was not clear, please let us know how we can improve it.
>
> ---
>
> > W3. Lack of ethical discussion: Despite the centrality of data ownership and provenance, the paper lacks a thorough discussion on potential misuse, legal concerns, or fairness implications of the proposed system.
>
> A. Given NeurIPS's technical focus, we prioritized addressing the computational challenges of enabling LLM-scale data valuation, as outlined in Appendix F.1. For readers interested in comprehensive discussions of AI data governance, including ethical and legal considerations, we recommend references [1, 2, 3] and will add these citations to Appendix F.1 to guide audiences seeking deeper exploration of these important topics.
>
> ---
>
> [1] Pahune et al., The importance of AI Data Governance in Large Language Models, 2024.
>
> [2] Reuel et al., Open Problems in Technical AI Governance, 2024.
>
> [3] Cheng et al., Training Data Attribution (TDA): Examining Its Adoption & Use Cases, 2025.

---

> ### Comment · Reviewer_fJkS · 2025-08-08
>
> Thank you for the reply.

---

### Official Review · Reviewer_mPa1 · 2025-07-02

**Clarity:** 3
**Significance:** 2
**Originality:** 2
**Rating:** 5
**Confidence:** 5

**Summary:**

This paper proposes an efficient data attribution algorithm by leveraging the gradient-projection approximation of the influence function with the special gradient structure for linear layers. Besides the algorithmic advancements, the authors also demonstrate how the algorithm can be efficiently implemented with the existing PyTorch library and provide a library that is capable of large-scale attribution.

**Questions:**

1. I have played around with the provided library, and I have some questions: In the `LoraLinear` class, why is the initialization of the projection matrix using `nn.init.kaiming_uniform_`? In addition, it seems like the implementation does not handle the biases in the linear layers.
2. (**Weakness 1**) Can the authors provide a rigorous statement of the Lemma and also the rigorous version of the proof? In particular, is it possible to avoid using $\approx$ in the statement (and also the proof)?
3. (**Weakness 3**) While I understand that the authors try to "propose" an overall logging/dot-product framework for data attribution, specifically the influence-function-style attribution, I think the key advancement lies in the introduction of the efficient gradient-projection in the logging stage. Hence, it is misleading to claim the speedups during the dot-product stage, especially when the baseline is largely limited by memory constraints. I suggest that the authors state the scenarios in which the speedup is obtained.

**Ethical Concerns:**

["NO or VERY MINOR ethics concerns only"]

**Final Justification:**

The authors addressed/answered all of my questions. In particular, the most important one being a potential over-claim on the speedup. For that, the authors also acknowledge the potential confusion, and promise to update to clarify this in the next iteration. Overall a technically solid paper that has a high impact on efficient gradient-based data attribution.

**Limitations:**

yes

**Quality:**

3

**Strengths And Weaknesses:**

**Strengths**
1. The research problem is well-motivated, and the scope is clear.
2. The overall writing and the illustration are professional and easy to follow.
3. The contribution is sound: in particular, the algorithmic design by exploiting the linear gradient structure, as well as the backward pass for the sample gradient calculation.
4. The evaluation is complete and thorough.

**Weaknesses**
1. The theory is not instructive and not entirely rigorous (in terms of the statement as well as the proof).
2. After some investigation, the structure of the codebase does not seem to be as extensible as claimed: the logic is scattered throughout the codebase, hard to track the lineage of definition for data/function.
3. The discussion around throughput gained is overclaimed: it comes from the dot products part, instead of the key technical advancements of this work, i.e., the logging part.

---

> ### Author Rebuttal · Authors · 2025-07-30
>
> Thanks for your detailed review on our work and code! Here, we attempt to address your questions:
>
> ---
>
> > Q1.1 I have played around with the provided library, and I have some questions: In the LoraLinear class, why is the initialization of the projection matrix using nn.init.kaiming_uniform_?
>
> A. We experimented with both uniform and normal initialization schemes for the projection matrices, but found no significant differences in our quantitative evaluations (regarding brittleness and LDS metrics). If you prefer to use different initialization strategies, you can easily modify it via `config.init_strategy`.
>
> > Q1.2 In addition, it seems like the implementation does not handle the biases in the linear layers.
>
> A. You're correct that we compute influence scores using only the weight parameters. Since weights account for the vast majority of network parameters, calculating influence scores based on weights alone typically provides sufficient accuracy. We'll clarify this experiment detail explicitly in our final manuscript to prevent the confusion for readers. Thanks for pointing it out!
>
> ---
>
> > Q2/W1. Can the authors provide a rigorous statement of the Lemma and also the rigorous version of the proof? In particular, is it possible to avoid using $\approx$ in the statement (and also the proof)?
>
> A. To our knowledge, approximating FIM with empirical FIM is frequently adopted in previous (inference function) literature [1, 2, 3, 4]. We used $\approx$ to explicitly indicate the approximation process, but if the reviewer thinks it only adds confusion, we can remove it in our equations in the final manuscript.
>
> ---
>
> > W2. After some investigation, the structure of the codebase does not seem to be as extensible as claimed: the logic is scattered throughout the codebase, hard to track the lineage of definition for data/function.
>
> A. We acknowledge that the codebase structure and function dependencies can be challenging to navigate. This complexity arose from our efforts to optimize logging performance and ensure compatibility with popular frameworks like Huggingface Transformers. Appendix E provides further details on the design considerations that shaped our implementation. When we claim extensibility, we're specifically referring to the statistics computation components (such as covariance, mean, and corrected eigenvalue calculations), which we anticipate will be the primary focus for practitioners looking to extend the framework. For instance, we expect fewer users will need to modify lower-level features like our memory-mapped file logging system.
>
> ---
>
> > Q3/W3. While I understand that the authors try to "propose" an overall logging/dot-product framework for data attribution, specifically the influence-function-style attribution, I think the key advancement lies in the introduction of the efficient gradient-projection in the logging stage. Hence, it is misleading to claim the speedups during the dot-product stage, especially when the baseline is largely limited by memory constraints. I suggest that the authors state the scenarios in which the speedup is obtained.
>
> A. We highlight the dot-product stage speedups because, in practical workflows, logging happens once while influence computations may be performed repeatedly across different queries (as noted in L86-90). This means the dot-product phase often becomes the primary bottleneck when practitioners explore multiple queries. Furthermore, we believe that the improvements in both stages are closely connected: the enhanced computational and memory efficiency during logging allows us to store comprehensive gradient information (e.g., gradient covariance and per-sample gradients), which in turn enables rapid influence calculations (dot-products) against all training examples. This end-to-end optimization is what makes our framework particularly effective for iterative data attribution analysis.
>
> ---
>
> Thanks again for your detailed feedback, and let me know if you have any further questions!
>
> [1] Kwon et al., DataInf: Efficiently Estimating Data Influence in LoRA-tuned LLMs and Diffusion Models. ICLR, 2024.
>
> [2] Wen et al., Interplay between optimization and generalization of stochastic gradient descent with covariance noise. AISTATS, 2020.
>
> [3] Chaudhari et al., Entropy-SGD: Biasing gradient descent into wide valleys. ICLR, 2017.
>
> [4] Kim et al., SqueezeLLM: Dense-and-Sparse Quantization. ICLR, 2024.

---

> ### Comment · Reviewer_mPa1 · 2025-08-01
>
> Thanks for the response. Overall I'm satisfied. Below are some of the follow-ups and feedbacks:
>
> > Q1.1+Q1.2
>
> I see. Thanks for the confirmation. In terms of the bias, I don't really have a big concern, just think it's something worth pointing out.
>
> > Q2/W1
>
> I think what I was suggesting is whether it is possible to derive some statistical bound on the approximation of Hessian ⇔ Empirical Hessian ⇔ FIM ⇔ Empirical FIM, and even Block-diagonal FIM. However, I admit that this might be difficult and out of reach right now.
>
> > W2
>
> Thanks for the clarification. I would like to point out that some recent work [1] around projection-based influence function has also explored other ways of implementing linear layer's gradient projection: they explicitly attach forward and backward hooks and project/reconstruct the projected gradients manually. I feel like while conceptually the LoGra structure is very elegant as it exploits the native gradient flow for modern autodiff library (e.g., adding an all-0 bottleneck layer, etc.), but at the first glance, manually handling these are actually cleaner and more straightforwards to me. Just want to bring this up as this seems relevant and you might be interested.
>
> [1]: GraSS: Scalable Influence Function with Sparse Gradient Compression
>
> > Q3/W3
>
> While I partially disagree with the claim that "in practice, dot-product phase often becomes the primary bottleneck" as the logging phase has a hard computational lower bound equivalent to one training epoch, I do agree that the proposed framework and also the key projection algorithms/implementations should be viewed together.
>
> I think a better way to put the result is to state the speed up in each stage respectively to better clarify the contribution.

---

> > ### Author Response · Authors · 2025-08-04
> > **Thanks for the response!**
> >
> > Thanks again for your very thoughtful feedback! Here, we leave short comments to your responses above. Please let us know if you have any further suggestions.
> >
> > ---
> >
> > > W2.
> >
> > We just checked the GraSS GitHub repo, and found that their code structure and our code structure in the supplemental share lots of similarities :) We agree that their hook-only implementation is also very elegant and straightforward. We will cite this paper in our final manuscript.
> >
> > ---
> >
> > > Q3/W3
> >
> > We agree that separately stating speed-ups from logging and influence computation stages in our abstract and introduction would better clarify the contribution. Thanks for the suggestion!

---

> ### Comment · Reviewer_mPa1 · 2025-08-04
>
> Thanks for the response for Q3/W3. I have no further follow-ups and/or concerns. Raising my rating to 5. Great work by the way, really enjoy it : )

---

> > ### Author Response · Authors · 2025-08-04
> >
> > We are really glad that you enjoyed our paper. Thank you so much!

---

### Official Review · Reviewer_BWig · 2025-07-03

**Clarity:** 4
**Significance:** 3
**Originality:** 4
**Rating:** 5
**Confidence:** 4

**Summary:**

This paper build on the use of influence functions to determine the relationship between model outputs and a vast set of training data. In particular the paper introduces an efficient method called LoGra that uses a low rank projection that can be efficiently integrated with the training process. The authors consider tradeoffs in the size and selection of the low rank projection (e.g. initialization through a PCA-like process). The method is first validated against competing methods in a small scale test using smaller models on typical datasets like CIFAR-10. The authors then test this method against a 1B token sized training set (a proxy used for access to actual training sets of models) and test realistic model LLM model sizes (i.e. billions of parameters). The authors show throughout where prior influence function methods may fail, often due infeasible computations and enormous systems costs.

**Questions:**

- Clarify the decisions around the 1B OWT data selection
- Clarify the usability of the software package
- Since qualitative comparisons (the example prompts) are part of the results for this method, are there possible experiments that could show that the data marked as valuable is of further use in model training? One version of this could use high value documents as ICL exemplars (or simply in context knowledge) to test the effects on evaluations.

**Ethical Concerns:**

["NO or VERY MINOR ethics concerns only"]

**Final Justification:**

I will maintain my positive score (5). The authors clarified some points in their response and I do not consider any of the raised weaknesses a reason to avoid acceptance.

**Limitations:**

Yes

**Paper Formatting Concerns:**

Typo:

315: when the queried LLM output itself is **(so?)** incoherent that its gradient does not encode meaningful information.

**Quality:**

4

**Strengths And Weaknesses:**

In general I found this paper to be very strong and convincing, with huge improvements over similar methods that allow this work to scale to both billion parameter data and models. While very dense, it is still clearly written and touches on a broad range of areas to show that their method is both theoretically sound and practically scalable.

## Strengths:

S1: The paper is very well motivated, discussing shortcomings in both "accuracy and efficiency" of existing systems (such as difficulties in computing solutions to the inverse hessian products). Continually considers practical limitations.

S2: Considers tweaks to the core method such as the PCA-like initialization of the low rank projections. The explanations here (3.2) motivate this idea and help understanding of the overall method.

S3: Considers systems aspects like maximizing available compute on typical GPU setups, recomputation, and storage costs of projected gradients.

S4: Gives some details about a software implementation of this method through the pytorch hooks interface. If this works as advertised (see W2), it makes this method even more relevant as it could be easily applied in many complex existing training stacks.

S5: Good baselines and comparisons of why some competing methods could not even be evaluated. I appreciate that the authors validate and compared the method on both small scale experiments similar to the literature and larger scales.

## Weaknesses

W1: I am a bit unclear on the selection of the 1B train tokens from OWT. This is meant to be a standin for direct actual access to a sample of the LLM training data, correct? Is it valid or accepted in data influence literature to use a proxy for training in this way? I believe these experiments could have used a subset of the dolma data and the olmo models, as these are both modern (i.e. compared to your issues with the pythia models) and open access. You note that OWT may be higher quality than the overal dolma mixture, but it also exists as high quality filtered subsets.

W2: Since the software package is considered a key contribution of this work (i.e. mentioned in the abstract) it would be good to show a practical end-to-end implementation of the package. Was the same package used to produce the LLM experiments?

W3: Could provide more details about the comment in 255-257 about complications ("... (i.e., no weight sharing) may not successfully keep larger components of the GPT2 Hessian...) with regards to "Transformer architecutre ... lacks the specialized KFAC Hessian approximation", as part of the central premise of the paper is that the method can be applied to billion param scale transformer models.

---

> ### Author Rebuttal · Authors · 2025-07-30
>
> Thanks for your reviewing effort and positive feedback! Here, we attempt to answer your questions.
>
> ---
>
> > Q. W1: I am a bit unclear on the selection of the 1B train tokens from OWT. This is meant to be a standin for direct actual access to a sample of the LLM training data, correct? Is it valid or accepted in data influence literature to use a proxy for training in this way? I believe these experiments could have used a subset of the dolma data and the olmo models, as these are both modern (i.e. compared to your issues with the pythia models) and open access. You note that OWT may be higher quality than the overal dolma mixture, but it also exists as high quality filtered subsets.
>
>
> A. We agree that OLMo + Dolma, similar to Pythia + Pile, is a valid experiment setup! For instance, C4 is a part of Dolma (similar to OWT being a part of Pile) that is frequently used in LLM pretraining, and we could have used C4 instead of OWT in our experiments. In our work, we chose one of a few reasonable data setups (i.e., OWT) and studied data valuation with multiple models (i.e., Pythia, GPT2, Llama3) to demonstrate the generality of the proposed method. While OWT may not be an exact subset of GPT2 and Llama3 pretraining data, we expect that OWT hugely overlaps with their pretraining data as mentioned in the paper, and thus the validity of our experiments largely hold. As for the common practice, we note that our work is one of very few work that studies data valuation at LLM pretraining scale with public models, so we hope that the more standardized experiment setup develops in the future.
>
> ---
>
> > Q. Since the software package is considered a key contribution of this work (i.e. mentioned in the abstract) it would be good to show a practical end-to-end implementation of the package. Was the same package used to produce the LLM experiments?
>
>
> A. All our experiments, including ones involving LLMs, are implemented with LogIX, and we provide the skeleton-code for our experiments in Appendix B. Furthermore, we included all our experiment code in the attached supplemental material. As for the implementation of LogIX itself, we described several important design principles and implementation details in Appendix F.
>
> ---
>
> > Q. Could provide more details about the comment in 255-257 about complications ("... (i.e., no weight sharing) may not successfully keep larger components of the GPT2 Hessian...) with regards to "Transformer architecutre ... lacks the specialized KFAC Hessian approximation", as part of the central premise of the paper is that the method can be applied to billion param scale transformer models.
>
>
> A. KFAC-based gradient compression is proposed in Section 3.2 as a technique to potentially improve accuracy of data valuation with LoGra. That being said, LoGra enables efficient data valuation with billion parameter-scale transformer models regardless of KFAC-based gradient compression, as empirically demonstrated in Section 4.1. We leave designing more accurate structured gradient compression techniques, especially for transformers, as interesting future work!
>
> ---
>
> > Q. Since qualitative comparisons (the example prompts) are part of the results for this method, are there possible experiments that could show that the data marked as valuable is of further use in model training? One version of this could use high value documents as ICL exemplars (or simply in context knowledge) to test the effects on evaluations.
>
>
> A. We would like to note that the brittleness experiment in Section 4.1 is closely connected to practical data curation tasks. For example, if the model shows an undesired behavior (e.g., gender/racial bias), we can trace this behavior back to its training data using influence functions and remove most influential data. Indeed, influence functions have a variety of applications including toxic dataset removal [1] and hallucination tracing [2]. Considering that the major bottleneck in using influence functions in those applications is the scalability, we focused on showcasing the scalability of LoGra in this paper.
>
> ---
>
> [1] Coalson, Z., Bae, J., Carlini, N., & Hong, S. (2025). $ IF-GUIDE $: Influence Function-Guided Detoxification of LLMs. arXiv preprint arXiv:2506.01790.
>
> [2] Wuhrmann, A., Kucherenko, A., & Kucharavy, A. (2025). Low-Perplexity LLM-Generated Sequences and Where To Find Them. arXiv preprint arXiv:2507.01844.

---

> > ### Comment · Reviewer_BWig · 2025-08-05
> >
> > Thanks for the detailed response! I'll maintain my positive score and congratulate the authors on a very interesting paper.

---

### Decision · Program_Chairs · 2025-09-17

**Decision:**

Accept (poster)

**Comment:**

The reviewers were strongly positive about this submission, agreeing that it tackles an important and timely problem with both technical depth and practical relevance. The proposed LoGra method makes influence-function–based data valuation feasible at LLM scale, and the accompanying LogIX toolkit lowers the barrier to adoption in real training pipelines. Reviewers emphasized that the work is well-motivated, carefully executed, and convincingly validated through experiments at a scale that prior approaches could not reach. They also noted that the paper is clearly written and likely to attract interest from both researchers and practitioners.

Some concerns were raised around dataset choice, overstatement of speedups, and the lack of a deeper ethical discussion, but the authors addressed these points thoughtfully in the rebuttal. After the exchange, reviewers either maintained or raised their scores, and there was broad agreement that the strengths; scalability, rigor, and practical utility, clearly outweigh the limitations. Overall, this is a technically solid and impactful piece of work, and I recommend acceptance.